

# 1 Paleo calendar-effect adjustments in time-slice and transient climate-model simulations (PaleoCalAdjust v1.0): impact and strategies for data analysis

Patrick J. Bartlein[1], Sarah L. Shafer[2]
[1]Department of Geography, University of Oregon, Eugene, OR 97403, USA
[2]Geosciences and Environmental Change Science Center, U.S. Geological Survey, Corvallis, OR 97331, USA
*Correspondence to*: Patrick J. Bartlein (bartlein@uoregon.edu)
**Abstract.** The "paleo calendar effect" is a common expression for the impact that the changes in the length of months or
seasons over time, related to changes in the eccentricity of Earth's orbit and precession, have on the analysis or summarization
of climate-model output. This effect can have significant implications for paleoclimate analyses. In particular, using a "fixed-
length" definition of months (i.e. defined by a fixed number of days), as opposed to a "fixed-angular" definition (i.e. defined
by a fixed number of degrees of the Earth's orbit), leads to comparisons of data from different positions along the Earth's orbit
when comparing paleo with modern simulations. This effect can impart characteristic spatial patterns or signals in comparisons
of time-slice simulations that otherwise might be interpreted in terms of specific paleoclimatic mechanisms, and we provide
examples for 6, 97, 116, and 127 ka. The calendar effect is exacerbated in transient climate simulations, where, in addition to
spatial or map-pattern effects, it can influence the apparent timing of extrema in individual time series and the characterization
of phase relationships among series. We outline an approach for adjusting paleo simulations that have been summarized using
a modern fixed-length definition of months and that can also be used for summarizing and comparing data archived as daily
data. We describe the implementation of this approach in a set of Fortran 90 programs and modules (PaleoCalAdjust v1.0).

## 20 1 Introduction

There are two ways of defining months or seasons (or any other portion of the year): 1) a "fixed-length" definition, where, for
example, months are defined by a fixed number of days (usually those at present), and 2) a "fixed-angular" definition, where,
again for example, months are defined by a fixed number of degrees of the Earth's orbit. Owing to the changes in Earth's orbit
over time these definitions will differ, and comparisons of paleo simulations with modern simulations using a fixed-length
definition of months will therefore incorporate data from different positions along the orbit, which can produce patterns in the
comparisons that mimic observed paleoclimatic changes. This paleo calendar effect arises from a consequence of Kepler's
second law of planetary motion: Earth moves faster along its elliptical orbit near perihelion, and slower near aphelion. Because
the time of year of perihelion and aphelion vary over time, the length of time that it takes the Earth to traverse one-quarter (90
degrees) or one-twelfth (30 degrees) of its orbit (a nominal season or month) also varies, so that months or seasons are shorter





near perihelion and longer near aphelion. For example, a 30- or 90-degree portion of the orbit will encompass a larger number
of days when the Earth is near perihelion (because it is moving faster along its orbit), and a smaller number when it is near
aphelion. Likewise, a 30- or 90-day interval will define a shorter orbital arc near perihelion, and a longer one near aphelion.
When examining present day and paleo simulations, summarizing data using a fixed-length definition of a particular month
(e.g. 31 days of a 365-day year), as opposed to a "fixed-angular" definition (e.g. (31 days / 365.25 days)•360 degrees of orbit,
where 365. 25 is the number of days in a year), will therefore result in comparing conditions that prevailed over different
portions of the Earth's orbit. Consequently, comparisons of present-day and paleoclimatic simulations that use the same
calendar (e.g. a present-day fixed-length calendar definition of January as 31-days long) will include two components of
change, one consisting of the actual model-simulated climate change between the present and paleo time period, and a second
arising simply from the difference in the angular portion of the orbit defined by 31 days at present as opposed to 31 days at the
paleo time period.
This impact of the calendar effect on the analysis of paleoclimatic simulations and their comparison with present-day or
"control" simulations is well known and not trivial (e.g. Kutzbach and Gallimore, 1988; Joussaume and Braconnot, 1997).
The effect is large and spatially variable, and can produce apparent map patterns that might otherwise be interpreted as evidence
of, for example, latitudinal amplification or damping of temperature changes, development of continental/marine temperature
contrasts, interhemispheric contrasts (the "bipolar seesaw"), changes in the latitude of the intertropical convergence zone
(ITCZ), variations in strength of the global monsoon, and others. In transient climate-model simulations, time series of data
aggregated using a fixed modern calendar, as opposed to an appropriately changing one, can differ not only in the overall shape
of long-term trends in the series, but also in variations in the timing of, for example, Holocene "thermal maxima" which,
depending on the time of year, can be on the order of several thousand years. The impact arises not only from the orbitally
controlled changes in insolation amount and the length of months or seasons, but also from the advancement or delay in the
starting and ending days of months or seasons relative to the solstices. Even if daily data are available, the calendar effect
must still be considered when summarizing those data by months or seasons, or when calculating climatic indices such as the
mean temperature of the warmest or coldest calendar month—values that are often used for comparisons with paleoclimatic
observations (e.g. Harrison et al., 2014, and see Kageyama et al., 2018, for further discussion). It is also the case that the
calendar effect can have a small impact on annual-average values, because the first day of the first month of the year may fall
in the previous year, and the last day of the last month of the year may fall in the next year.
Various approaches have been proposed for incorporating the calendar effect or "adjusting" monthly values in analyses of
paleoclimatic simulations (e.g. Pollard and Reusch, 2002; Timm et al., 2008; Chen et al., 2011). Despite this work, the calendar
effect is generally ignored, and so our motivation here is to provide an adjustment method that is relatively simple and can be
applied generally to "CMIP-formatted" (https://esgf-node.llnl.gov/projects/cmip5/) files, such as those distributed by the
Paleoclimate Modelling Intercomparison Project (PMIP, Kageyama et al., 2018). Our approach (broadly similar to Pollard



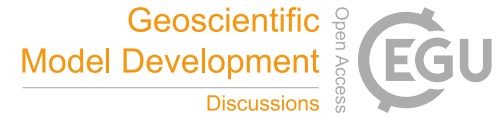

and Reusch, 2002) involves (1) determining the appropriate fixed-angular month lengths for a paleo experiment (e.g., Kutzbach
and Gallimore, 1988), (2) interpolating the data to a daily time step using a mean-preserving interpolation method (e.g.,
Epstein, 1991), and then (3) averaging or accumulating the interpolated daily data using the appropriate (paleo) month starting
and ending days, thereby explicitly incorporating the changing month lengths. In cases where daily data are available (e.g. in
CMIP5/PMIP3 "day" files), only the third step is necessary. This approach is implemented in a set of Fortran 90 programs
and modules (PaleoCalAdj v1.0, described below). With a suitable program code "wrapper" file, the approach can also be
applied to transient simulations (e.g. Liu et al., 2009; Ivanovic et al., 2016).
In the following discussion, we describe (a) the calendar effect on month lengths and their beginning, middle and ending days
over the past 150 kyr; (b) the spatial patterns of the calendar effect on temperature and precipitation rate for several key times
(6, 97, 116, and 127 ka); and (c) the methods that can be used to calculate month lengths (on various calendars) and to "calendar
adjust" monthly or daily paleo model output to an appropriate paleo calendar.
**2 Month-length variations**
The fixed-angular length of months as they vary over time can be calculated using the algorithm in Appendix A of Kutzbach
and Gallimore (1988). This algorithm yields the length of time (in real-number or fractional days) required to traverse a given
number of degrees of celestial (as opposed to geographical) longitude starting from the vernal equinox, the common "origin"
for orbital calculations (see Jousaumme and Braconnot, 1997, for discussion). Although developed for a 360-day year with
30-day months at present, the algorithm can easily be adapted for other calendars and for cases where the present-day months
are not equal in length. The beginnings and ends of each fixed-angular month in a 365-day "noleap" calendar are shown at 1
kyr intervals for the past 150 kyr in Fig. 1, calculated using the approach described in Sects. 4.2-4.5 below. (See section 4.4.1
of the *NetCDF Climate and Forecast Metadata Conventions* (http://cfconventions.org/) for a discussion of climate-model
output calendar types.) The month-length "anomalies" (i.e. long-term differences between paleo and present month lengths,
with present defined as 1950 CE) are shown in color, with (paleo) months that are shorter than those at present in green shades,
and months that are longer than those at present in blue shades. Not only do the lengths of fixed-angular months vary over
time, but so do their middle, beginning and ending days (Fig. 2), with mid-month days that are closer to the June solstice
indicated in orange and those that are farther from the June solstice in blue. The variations in month length (Fig. 1) obviously
track the changing time of year of perihelion, while the beginning and ending day anomalies reflect the climatic precession
parameter (Fig. 2). The shift in the beginning, middle, or end of individual months relative to the solstices ultimately controls
the average or mid-month daily insolation at different latitudes (Figs. 3-5). The calendar effect is illustrated below for four
times: 6 and 127 ka are the target times for the planned warm-interval *midHolocene* and *lig127k* CMIP6/PMIP4 (Coupled
Model Intercomparison Project Phase 6/Paleoclimate Modelling Intercomparison Project Phase 4) simulations (Otto-Bliesner
et al., 2017) and illustrate the calendar effects when perihelion occurs in the boreal summer or autumn (Fig. 6); 116 ka is the



time of a proposed sensitivity experiment for the onset of glaciation (Otto-Bliesner et al., 2017), and illustrates the calendar
effect when perihelion occurs in boreal winter; and 97 ka was chosen to illustrate an orbital configuration not represented by
the other times (i.e. one with boreal spring months occurring closer to the June solstice).
At 6 ka, perihelion occurred in September (Fig. 6), and the months from May through October were shorter than today (Fig.
1), with the greatest differences in August (1.65 days shorter than present).  This contraction of month lengths moved the
middle of all of the months from April through December closer to the June solstice (Fig. 2), with the greatest difference in
November (4.9 days closer to the June solstice, and so 4.9 days farther from the December solstice).  At 127 ka, perihelion
was in late June, and the months April through September were shorter than today (Fig. 1), with the greatest difference in July
(3.23 days shorter than present).  As at 6 ka, the shorter boreal summer months move the middle of the months between July
and December closer to the June solstice (Fig. 2), with the greatest difference in September and October (12.4 and 12.3 days
closer, respectively).  At both 6 and 127 ka, the longer boreal winter months begin and end earlier in the year, placing the
middle of January 3.3 (6 ka) and 4.3 (127 ka) days farther from the June solstice than at present.  As can be noted on Figs. 1
and 2, 127 ka does not represent a simple amplification of 6 ka conditions.  Although broadly similar in having shorter late
boreal summer and autumn months that begin earlier in the year (and hence closer to the June solstice), the two times are only
similar in the relative differences from present in month length and beginning and ending days.
At 116 ka, perihelion was in late December, and consequently the months from October through March were shorter than
present (Fig. 1).  This has the main effect of moving the middle of the months July through December farther from the June
solstice (with a maximum in September of 5.6 days; Fig. 2), somewhat opposite to the pattern at 6 and 127 ka.  At 97 ka,
perihelion occurred in mid-November, in between its occurrence in September at 6 ka and December at 116 ka (Fig. 1).  The
impact on month length and mid-month timing is complicated, with the mid-month days of January through March and July
through October occurring farther from the June solstice (Fig. 2).
The first-order impact of the calendar effect can be gauged by comparing (at a particular latitude) daily insolation values for
mid-month days determined using the appropriate paleo calendar (which assumes fixed-angular definitions of months) with
insolation values for mid-month days using the present-day calendar (which assumes fixed-length definitions of months).  At
6 ka, at 45° N, the shorter (than present), and earlier (relative to the June solstice) months of September through November
had insolation values over 10 W m$^{-2}$ greater for mid-month days defined using the fixed-angular paleo calendar, in comparison
with values determined using the fixed-length present-day calendar (Fig. 3), and at 127 ka, the differences exceeded 35 W m$^{-2}$
for the months of August through October.  These positive insolation differences were accompanied by negative differences
from January through June.  At first glance, it may be counterintuitive that the calendar effects that yield positive differences
in mid-month insolation are not balanced by the negative insolation differences as is the case with the month-length differences.
However, the calendar effects on insolation include both the month-length differences as well as long-term insolation





differences themselves (Figs. 7-9), which are not symmetrical within the year, and so the calendar effects do not "cancel out"
within the year.
At 116 ka, the longer but later occurring month of September had a negative difference in mid-month insolation that exceeded
10 W m$^{-2}$. For regions where surface temperatures are strongly tied to insolation with little lag, such as the interiors of the
northern continents, these calendar effects on insolation will directly be reflected by the calendar effects on temperatures. By
moving the beginning, middle and end of individual months (and seasons) closer to or farther from the solstices, the "apparent
temperature" of those intervals will be affected (i.e. months or seasons that start or end closer to the summer solstice will be
warmer). The calendar effect on insolation varies strongly with latitude, with the sign of the difference broadly reversing in
the southern hemisphere (Figs. 3-5).
**3 Impact of the calendar effect**
Past demonstrations of the calendar effect have used "real" paleoclimatic simulations, and so the climate patterns being used
in these demonstrations include both the calendar effect, and the long-term mean differences in climate between experiment
and control simulations. Comparison of Figs. 3 and 7 clearly shows, however, that the variations over time in insolation and
in the calendar effect are not identical, and so the use of an actual paleoclimatic experiment (e.g. for 6 ka or 127 ka) to illustrate
the calendar effect will inevitably be confounded by the climatic response to changes in insolation (and other boundary
conditions). The impact on the analysis of paleoclimatic simulations of the calendar effect can alternatively be assessed by
assuming that the long-term mean difference in climate (also referred to as the experiment minus control "anomaly") is zero
everywhere, illustrating the "pure" calendar effect. Pseudo-daily interpolated values (or actual daily output, if available) of
present-day monthly data can then simply be reaggregated using an appropriate paleo calendar and compared with the present-
day data. (The pseudo-daily values used here were obtained by interpolating monthly data to a daily time-step using the
monthly mean-preserving algorithm described below.)
The "pure" calendar effect is demonstrated here using present-day monthly long-term mean (1981-2010) values of near-surface
air temperature (*tas*) from the Climate Forecast System Reanalysis (CFSR; Saha et al., 2010;
https://esgf.nccs.nasa.gov/projects/ana4mips/), and monthly precipitation rate (*precip*) from the CPC Merged Analysis of
Precipitation (CMAP; Xie and Arkin, 1997; https://www.esrl.noaa.gov/psd/data/gridded/data.cmap.html) (Fig. 10). (These
data were chosen because they are global in extent and are of reasonably high spatial resolution.) If it is assumed that there is
no long-term mean difference between a present-day and paleo simulation (by adopting the present-day data as the simulated
paleo data), then the unadjusted present-day data can be compared with present-day data adjusted to the appropriate paleo
month lengths. The calendar-adjusted minus unadjusted differences will therefore reveal the inverse of the built-in calendar
effect "signal" in the unadjusted data, that might readily be interpreted in terms of some specific paleoclimatic mechanisms,



while being instead a data analytical artefact. Positive values on the maps (Figs. 11-13) indicate, for example, where
temperatures would be higher or precipitation greater if a fixed-angular calendar were used to summarize the paleo data.

## 3.1 Monthly temperature

The impacts of using the appropriate calendar to summarize the data (as opposed to not) are large, often exceeding 1 °C in
absolute value (Fig. 11). The effects are spatially variable, and are not simple functions of latitude as might be initially
expected, because the effect increases with the amplitude of the annual cycle (which has a substantial longitudinal component)
for temperature regimes that are in phase with the annual cycle of insolation. For temperature regimes that are out of phase
with insolation, the calendar-adjusted minus unadjusted values would be negative, and largest when the temperature variations
were exactly out of phase. (If there were no annual cycle, i.e. if a climate variable remained constant over the course of a year,
the calendar effect would be zero.) The interaction between the annual cycle and the direct calendar effect on insolation
produces patterns of the overall calendar effect that happen to resemble some of the large-scale responses that are frequently
found in climate simulations, both past and future, such as high-latitude amplification or damping, continental-ocean contrasts,
interhemispheric contrasts and changes in seasonality of temperature (cf. Izumi et al., 2013). Because the month-length
calculations use the Northern Hemisphere vernal equinox as a fixed origin for the location of Earth along its orbit, the effects
seem to be small during the months surrounding the equinox (i.e. February through April, Fig. 11), and indeed the selection of
a different origin would produce different apparent effects (see Jousaumme and Braconnot, 1997, Sect. 2.1). However, the
selection of a different origin would not change the relative (to present) length of time it would take Earth to transit any
particular angular segment of its orbit.
At 6 ka, the largest calendar effects on temperature can be observed over the Northern Hemisphere continents for the months
from September through December (Fig. 11), consistent with the earlier beginning of these months (Fig. 2) and the direct
calendar effect on insolation at 45° N (Fig. 3). For example, in the interior of the northern continents, as well as North Africa,
temperature is in phase with insolation, and so the calendar effect on insolation (Fig. 3), which produces strongly positive
differences from August through November, is reflected by the calendar effect on temperature. Over the northern oceans,
temperature is broadly in phase with insolation, but with a lag, which reduces the magnitude of the effect and gives rise to an
apparent land-ocean contrast that otherwise might be interpreted in terms of some particular paleoclimatic mechanism. The
calendar effect on temperature from January through March produces negative calendar-adjusted minus unadjusted values in
the northern continental interiors (Fig. 11), which is also consistent with the calendar effect on insolation. In the Southern
Hemisphere at 6 ka, the calendar effects on temperature produce generally negative differences, which is consistent with the
calendar effects on mid-month insolation at 45° S (Fig. 5), which produce generally negative differences throughout the year,
particularly during the months of August through November. Like the continent – ocean contrast in the Northern Hemisphere,
the Northern Hemisphere – Southern Hemisphere contrast in the calendar effect on temperature also could be interpreted in
terms of one or another of the mechanisms thought to be responsible for interhemispheric temperature contrasts.





At 127 ka, the calendar effect on temperature is broadly similar to that at 6 ka over the months from September through March,
but differs in sign from April through July, and in magnitude in August (Fig. 11).  These patterns are also consistent with the
direct calendar effects on insolation.  At 127 ka, the calendar effect on insolation produces strongly positive differences in the
Northern Hemisphere earlier in the northern summer than at 6 ka (Fig. 3), while at 45° S the calendar effect on insolation
produces strongly negative differences in July and persists that way through November (Fig. 5).  At 116 ka, perihelion occurs
in late December, in comparison to late June at 127 ka (Figs. 1 and 6), and not surprisingly the calendar effect on temperature
is nearly the inverse of that at 127 ka (Fig. 11).  This pattern has important implications for paleoclimatic studies, because in
addition to all of the changes in the forcing and the paleoclimatic responses accompanying the transition out of the last
interglacial, the possibility that some of the apparent simulated changes between 127 and 116 ka may be an artefact of data-
analysis procedures cannot be discounted.
At 97 ka, a time selected to illustrate a different orbital configuration (i.e. one with boreal spring months occurring closer to
the June solstice) than the similar (6 ka and 127 ka) or contrasting (127 and 116 ka) configurations, the calendar effect on
temperature in the Northern Hemisphere (Fig. 11) shows a switch from positive differences in the early boreal summer (May
and June) to negative in the late summer (August and September).  This switch is again consistent with the direct calendar
effect on insolation (Fig. 3).  Like the other times, the spatial variations in the calendar effect could easily be interpreted in
terms of one kind of paleoclimatic mechanism or another.

## 3.2 Mean temperature of the warmest and coldest months

Although the calendar effects on monthly mean temperature show some sub-continental scale variability, the overall patterns
are of relatively large spatial scales, and are interpretable in terms of the direct orbital effects on month lengths and insolation.
The calendar effects on the mean temperature of the warmest (MTWA) and coldest (MTCO) calendar months (and their
differences) are much more spatially variable (Fig. 12).  This variability arises in large part because of the way these variables
are usually defined (e.g. as the mean temperature of the warmest or coldest conventionally defined month, as opposed to the
temperature of the warmest or coldest 30-day interval), but also because the calendar adjustment can result in a change in the
specific month that is warmest or coldest.  These effects are compounded when calculating seasonality (as MTWA minus
MTCO).  Other definitions of the warmest and coldest month are possible, such as the warmest consecutive 30-day period
during the year (e.g. Caley et al., 2014), and such definitions will not be susceptible to the calendar effect.  In practice, however,
paleoclimatic reconstructions based on calibrations or forward-model simulations routinely use conventional calendar-month
definitions of the warmest and coldest months and of seasonality (Bartlein et al., 2011; Harrison et al., 2014), and often only
monthly output from paleoclimatic simulations is available necessitating consistent definitions when summarizing model
output.





In the particular set of example times chosen here, the magnitudes of the calendar effects are also smaller than those of
individual months because, as it happens, the calendar effects in January and February (typically coldest months in the Northern
Hemisphere) and July and August (typically warmest months in the Northern Hemisphere) are not large. There are also some
surprising patterns. The inverse relationship between the calendar effects at 116 ka and 127 ka that might be expected from
inspection of the monthly effects (Fig. 11) are not present, while the calendar effects on MTCO and MTWA at 97 ka and 116
ka tend to resemble one another. Across the four example times, there is an indistinct, but still noticeable pattern in reduced
seasonality (MTWA minus MTCO) between the adjusted and unadjusted values, which like the other patterns described above
could tempt interpretation in terms of some specific climatic mechanisms.

### 3.3 Monthly precipitation

In contrast to the large spatial-scale patterns of the calendar effect on temperature, the patterns of the calendar effect on
precipitation rate are much more complex, showing both continental-scale patterns (like those for temperature), but also
smaller-scale patterns that are apparently related to precipitation associated with the ITCZ and regional and global monsoons
(Fig. 13). The continental-scale patterns are evident in the calendar effects at 6 and 127 ka, particularly in the months from
September through November (Fig. 13), where it also can be noted (especially over the mid-latitude continents in both
hemispheres) that there is a positive association with the calendar effect on temperature. This association is related simply to
similarities in the shapes of the annual cycles of those variables, and not to some kind of more elaborate thermodynamic
constraint. At 116 ka, as for temperature, the large-scale calendar-effect patterns appear to be nearly the inverse of those at
127 ka. The smaller-scale kind of pattern is well illustrated at 127 ka in the tropical North Atlantic, sub-Saharan Africa and
south Asia. There, negative calendar-adjusted minus unadjusted values can be noted for June through August, giving way to
positive differences from September through November, and the same transition appears inversely at 116 ka. Another example
can be found in the South Pacific Convergence Zone in austral spring and early summer (September through November) at 6
and 127 ka, where generally positive differences between calendar-adjusted and unadjusted values in July and August gives
way to negative differences from September through December. This second kind of pattern, most evident in the subtropics,
is not mirrored by the calendar effects on temperature.
Overall, the magnitude and spatial patterns of the calendar effects on temperature and precipitation (Figs. 11 and 13) resemble
those in the paleoclimatic simulations and observations that we attempt to explain in mechanistic terms. Depending on the
sign of the effect, neglecting to account for the calendar effects could spuriously amplify some "signals" in long-term mean
differences between experiment and control simulations, while damping others.

### 3.4 Calendar effects and transient experiments

Calendar effects must also be considered in the analysis of transient climate-model simulations (even if those data are available
on the daily time step). This can be illustrated for a variety of variables and regions using data from the TraCE-21k transient



simulations (Liu et al., 2009; https://www.earthsystemgrid.org/project/trace.html). The series plotted in Fig. 14 are area-
averages for individual months on a yearly time step, with 100-yr (window half-width) locally weighted regression curves
added to emphasize century-timescale variations. The original yearly time-step data were aggregated using a perpetual "no
leap" (365-day) calendar (using the present-day month lengths for all years). The gray and black curves on Fig. 14 show these
unadjusted "original" values, while the colored curves show month-length adjusted values (i.e. pseudo-daily interpolated
values, reaggregated using the appropriate paleo fixed-angular calendar). Area averages were calculated for ice-free land
points.
Figure 14a shows area-weighted averages for 2 m air temperature for a region that spans 15 to 75° N and -170 to 60° E, the
region used by Marsicek et al. (2018) to discuss Holocene temperature trends in simulations and reconstructions. The largest
differences between month-length adjusted values and unadjusted values occur in October between 14 and 6 ka, when
perihelion occurred during the northern summer months. October month lengths during this interval were generally within
one day of those at present (Fig. 1), but the generally shorter months from April through September resulted in Octobers
beginning up to ten days earlier in the calendar than at present, i.e. closer in time to the boreal summer solstice (Fig. 2). The
calendar-effect adjusted October values therefore average up to 4 °C higher than the unadjusted values during this interval
(Fig. 14a), consistent with the direct calendar effects on insolation at 45° N (Fig. 3). The calendar effect also changes the
shape of the temporal trends in the data, particularly during the Holocene. October temperatures in the unadjusted data showed
a generally increasing trend over the Holocene (i.e. since 11.7 ka), reaching a maximum around 3 ka, comparable with present-
day values, while the adjusted data reached levels consistently above present-day values by 7.5 ka. The unadjusted October
temperature data could be described as reaching a "Holocene thermal maximum" only in the late Holocene (i.e., after 4 ka),
while the adjusted data display more of a mid-Holocene maximum. As is the case with the mapped assessments of the "pure"
calendar effect, the differences between unadjusted and adjusted time series are of the kind that could be interpreted in terms
of various hypothetical mechanisms. For example, the calendar-effect adjustment advances the time of occurrence of a
Holocene thermal maximum in October by about 3 kyr for North America and Europe.
As in North America and Europe, the adjusted temperature trends in Australia (10 to 50° S and 110 to 160° E) (Fig. 14b) are
consistent with the direct calendar effects on insolation (i.e. for 45° S, Fig. 5). The difference between adjusted and unadjusted
values are again largest in October between 14 and 6 ka, but the difference is the inverse of that for the North America and
Europe region, because the annual cycle of temperature for Australia is inversely related to the annual cycle of the insolation
anomalies (Fig. 9) and so to the direct calendar effects on insolation (Fig. 5). Again, the shapes of the Holocene trends in the
adjusted and unadjusted data are noticeably different. In the Australia (Fig. 14b) and North America and Europe (Fig. 14a)
examples, relatively large areas are being averaged, and the calendar effect becomes more apparent as the size of the area
decreases. Notably, the effect does not completely disappear at the largest scales, i.e. for area-weighted averages for the globe
(for ice-free land grid cells) (Fig. 14c). The differences are smaller, but still discernible.



In the Northern Hemisphere (African-Asian) Monsoon region (0 to 30° N and -30 to 120° E), the calendar effects on precipitation rate are similar to those on temperature in the mid-latitudes because the annual cycle of precipitation is roughly in phase with that of insolation (Fig. 7). There is little effect in the winter and spring, but a substantial effect in summer and autumn over the interval from 17 ka to about 3 ka (Fig. 14d). The calendar effect reverses sign between July and August (when the month-length adjusted precipitation rate values are less than the unadjusted ones) and September and October (when the adjusted values are greater than the unadjusted ones). In July, the timing of relative maxima and minima in the two data sets is similar, while in October, in particular, the Holocene precipitation maximum is several thousand years earlier in the adjusted data than in the unadjusted.

The time-series expression of the latitudinally reversing calendar effect on precipitation rate evident in Fig. 13 (e.g. July vs. October at 127 ka) can be illustrated by comparing precipitation or precipitation minus evaporation (*P* - *E*) for the North African (sub-Saharan) Monsoon region (5 to 17° N and -5 to 30° E) with the Mediterranean region (31 to 43° N and -5 to 30° E) (Fig. 14e and 14f). The differences between the adjusted and unadjusted data in the North African region (Fig. 14e) parallel that of the larger monsoon region (Fig. 14d). The Mediterranean region, which is characteristically moister in winter and drier in summer shows the reverse pattern: when the calendar-adjusted minus unadjusted *P* - *E* difference is positive in the monsoon region, it is negative in the Mediterranean region. Dipoles are frequently observed in climatic data, both present-day and paleo, and are usually interpreted in terms of broad-scale circulation changes in the atmosphere or ocean. This example illustrates that they could also be artefacts of the calendar effect. Such changes in timing of extrema also could influence the interpretation of phase relationships among simulated time series and time series of potential forcing (Joussaume and Braconnot, 1997; Timm et al., 2008; Chen et al., 2011). The impacts of the calendar effect on temporal trends in transient simulations (Fig. 14), when compounded by the spatial effects (Figs. 11-13), make it even more likely spurious climatic mechanisms could be inferred in analyzing transient simulations than in the simpler time-slice simulations.

## 4 PaleoCalAdjust v 1.0

The approach we describe here for adjusting model output reported either as monthly data (using fixed-length definitions of months) or as daily data to reflect the calendar effect (i.e. to make month-length adjustments) has two fundamental steps: 1) pseudo-daily interpolation of the monthly data on a fixed-month-length calendar (which, when actual daily data are available, is not necessary), followed by 2) aggregation of those daily data to fixed-angular months defined for the particular time of the simulations. The second step obviously requires the calculation of the beginning and ending days of each month as they vary over ("geological") time, which in turn depends on the orbital parameters. The definition of the beginning and ending days of a month in a "leap-year," "Gregorian," or "proleptic Gregorian" calendar (http://cfconventions.org) additionally depends on the timing of the (northern) vernal equinox, which varies from year to year. Here we describe the pseudo-daily interpolation method first, followed by a discussion of the month-length calculations. Then we describe the calendar-adjustment program,



along with a few demonstration programs that exercise some of the individual procedures. All of the programs, written in
Fortran 90, are available (see *Code and data availability* section).

## 4.1 Pseudo-daily interpolation

The first step in adjusting monthly time-step model output to reflect the calendar effect is to interpolate the monthly data (either
long-term means or time-series data) to pseudo-daily values. (A step that is not required if the data are daily time-step values.)
It turns out that the most common way of producing pseudo-daily values, linear interpolation between monthly means, is not
mean preserving; the monthly (or annual) means of the interpolated daily values will generally not match the original monthly
values. An alternative approach, and the one we use here, is the mean-preserving "harmonic" interpolation method of Epstein
(1991), which is easy to implement, and performs the same function as the parabolic-spline interpolation method of Pollard
and Reusch (2002).
The linear and mean-preserving interpolation methods can be compared using the Climate Forecast System Reanalysis (CFSR)
near-surface air temperature and CPC Merged Analysis of Precipitation (CMAP) 1981-2010 long-term mean data (Fig. 15).
A typical example for temperature appears in Fig. 15a, for a gridpoint near Madison, Wisconsin (USA). The difference
between the annual mean values of the interpolated data for the two approaches is small and similar (ca. $2.0 \times 10^{-6}$), but the
difference between the original monthly means and the monthly mean of the linearly interpolated daily values can exceed 0.8
°C in some months (e.g. December). (The differences from the original monthly means for the mean-preserving interpolation
method are less than $1.0 \times 10^{-3}$ °C for every month in Fig. 15a.) Fig. 15b shows an example for a grid point in Australia, where
again the difference between the original monthly means and the monthly means of the linearly interpolated daily values is not
negligible (i.e. 0.4 °C). Similar results hold for precipitation (Fig. 15c), where the difference can exceed 0.1 mm $d^{-1}$). Like
other harmonic-based approaches, the Epstein approach can create interpolated curves that are wavy (see Pollard and Reusch
(2002) for discussion), but these effects are small enough to not be practically important in nearly all cases. The pathological
case for precipitation is shown in Fig. 15d, at a grid point in the Indian Ocean. Here, the difference between an original
monthly mean value and one calculated using the mean-preserving interpolation method reaches -0.12 mm $d^{-1}$ in March and
April, but the differences between the original monthly means and the monthly means of the linearly interpolated daily values
are nearly three times larger.
The map patterns of the interpolation errors (the monthly mean values recalculated using the pseudo-daily interpolated values
minus the original values) appear in Fig. 16. (Note the differing scales for the linear-interpolation errors and the mean-
preserving-interpolation errors.) The linear interpolation errors are quite large, with absolute values exceeding 1 °C and 1 mm
$d^{-1}$, and have distinct seasonal and spatial patterns: underpredictions of Northern Hemisphere temperature in summer (and
overpredictions in winter), and underpredictions of precipitation in the wet season (e.g. southern Asia in July) and
overpredictions in the dry season (southern Asia in May). The magnitude and patterns of these effects again rival those we




attempt to infer or interpret in the paleo record. The mean-preserving interpolation errors for temperature are very small, and
show only vague spatial patterns (note the differing scales). The errors for precipitation are also quite small, but can be locally
larger, as in the pathological case illustrated above. However, the map patterns of the interpolation errors strongly suggest that
those cases are not practically important.
The mean-preserving interpolation method is implemented in the Fortran 90 module named `pseudo_daily_interp_subs.f90`.
The subroutine `hdaily(…)` manages the interpolation, first getting the harmonic coefficients (Eq. 6 of Epstein, 1991) using the
subroutine named `harmonic_coeffs(…)` and then applying these coefficients in the subroutine `xdhat(…)` to get the interpolated
values.
**4.2 Month-length calculations**
Calculation of the length and the beginning, middle and ending (real-number or fractional) days of each month at a particular
time is based on the algorithm described by Kutzbach and Gallimore (1988, see also Kutzbach and Otto-Bliesner, 1982). The
algorithm allows the calculation of the length of time, in real-number or fractional days, for the Earth to traverse an angular
portion of its orbit, i.e. it provides the length of months or seasons in a fixed-angular calendar for any particular time. By
choosing a fixed reference day (i.e. the vernal equinox), the beginning, middle, and ending days of months can be calculated.
Application of this algorithm requires as input eccentricity and the longitude of perihelion (in degrees) relative to the vernal
equinox, and the generalization of the approach to other calendars, such as the "proleptic Gregorian" calendar (that includes
leap years, http://cfconventions.org), also requires the (real-number or fractional) day of the vernal equinox. To calculate the
orbital parameters using the Berger (1978) solution, and the timing of the (northern) vernal equinox (as well as insolation
itself), we adapted a set of programs provided by National Aeronautics and Space Administration, Goddard Institute for Space
Studies (https://data.giss.nasa.gov/ar5/solar.html).
The approach adopted by Kutzbach and Gallimore (1988) is based on an approximation that describes the rate of change in
celestial longitude, $\phi$, with time (over the year):
$\qquad dt/d\phi = 1 + 2e \sin((2\pi/360)(\phi - \phi_P))$ (1)
which depends on eccentricity, $e$, and the date of perihelion, expressed as a phase angle, $\phi_P$, defined so that $\sin((2\pi/360)(\phi -$
$\phi_P)) = -1$ at the celestial longitude of perihelion, and where $\phi$, and $\phi_P$ are expressed in units of degrees and $t$ in days (their
equation A1). After $\phi_P$ has been determined, the amount of time (in real-number or fractional days) required to traverse a given
number of degrees of celestial longitude from the vernal equinox can be determined by an integration of A1 (their equation
A2):
$\qquad t = \phi - 2e (\cos((2\pi/360)(\phi - \phi_P)) - \cos((2\pi/360)\phi))(360/2\pi)$ (2)
We implemented this approach in the subroutine `kg_monlen_360(…)` in the Fortran 90 module named `month_length_subs.f90`.
(This subroutine is not actually used in practice because it can handle only 360-day year calendars, but it illustrates the basic



ideas of the approach.)  After initializing a set of day numbers and angular differences from the vernal equinox (assumed to be fixed at 80 days after the beginning of the year) (Step 1 in `kg_monlen_360(…)`), we determine $\phi_p$ by advancing along the orbit at 0.001-day increments from the vernal equinox, and selecting $\phi_p$ as the value that minimizes $-1 - \sin((2\pi/360)(\phi - \phi_p))$ (Step 2).  Then the traverse time since the vernal equinox is calculated for each day using Kutzbach and Gallimore's (1988) equation A2 (Step 3), and this traverse time is used to get the relative length of each day through simple differencing (Step 4). Finally, the length of each month (in real-number or fractional days) is determined by accumulation (Step 5).

**4.3 Simulation ages and simulation years**

Inspection shows that different climate models employ different starting dates in their output files for both present-day (*piControl*) and paleo (e.g. *midHolocene*) simulations (https://esgf-node.llnl.gov/projects/cmip5/).  For models that use a noleap (constant 365-day year) calendar, such as CCSM4 (Otto-Bliesner, 2014), the starting date is not an issue, but for MPI-ESM-P (Jungclaus et al., 2012), which uses a proleptic Gregorian calendar, or CNRM-CM5 (Sénési et al., 2014), with a "standard" (i.e. mixed Julian/Gregorian) calendar as examples, the specific starting date influences the date of the vernal equinox through the occurrence of individual leap years.  For example, in the CMIP5/PMIP4 *midHolocene* simulations, output from MPI-ESM-P starts in 1850 CE, and that from CNRM-CM5 in 2050 CE (and it can be verified that leap years in those output files occur in a fashion consistent with the "modern" calendar).  Consequently, we need to make a distinction between two notions of time here: 1) the simulation age, expressed in (negative) years BP (i.e. before 1950 CE), and 2) the simulation year, expressed in years CE.  The simulation age controls the orbital parameter values, while the simulation year, along with the specification of the CF-compliant calendar attribute (http://cfconventions.org), controls the date and time of the vernal equinox.

**4.4 Month-length programs and subprograms**

Month lengths are calculated in the subroutine, `get_month_lengths(…)` (contained in the Fortran 90 module named `month_length_subs.f90`), that in turn calls the subroutine `kg_monlen(…)` to get real-type month lengths for a particular simulation age and year. (The subroutine `get_month_lengths(…)` can be exercised to produce tables of month lengths, beginning, middle and ending days of the kind used to produce Figs. 1-5 and 7-9 using a driver program named `month_length.f90`.)  The subroutine `get_month_lengths(…)` uses two other modules, `GISS_orbpar_subs.f90` and `GISS_srevents_subs.f90` (based on programs downloaded from GISS (https://data.giss.nasa.gov/ar5/solar.html)), to get the orbital parameters and vernal equinox dates.

The specific tasks involved in the calculation of either a single year's set of month lengths, or a series of month lengths for multiple years, include the following steps, implemented in `get_month_lengths(…)`:

1. generate a set of "target" dates based on the simulation ages and simulation years;



2.  obtain the orbital parameters for 0 ka (1950 CE), which will be used to adjust the calculated month-length values to the conventional definition of months for 1950 CE as the reference year;

3.  obtain the present-day (i.e. 1950 CE) month lengths for the appropriate calendar.

Then loop over the simulation ages and simulation years, and for each combination:

4.  obtain the orbital parameters for each simulation age, using the subroutine `GISS_orbpars(…)`;

5.  calculate real-type month lengths for the appropriate calendar using `kg_monlen(…)`;

6.  adjust (using the subroutine `adjust_to_reference(…)`) those month length values to the reference year (e.g. 1950 CE) and its conventional set of month-length definitions so that, for example, January will have 31 days, February 28 or 29 days, etc., in that reference year;

7.  further adjust the month-length values to ensure that the individual monthly values will sum exactly to the year length in days using `adjust_to_yeartot(…)`;

8.  convert real-type month lengths to integers using `integer_monlen(…)` (These integer values are not used anywhere, but may be useful in conceptualizing the pattern of month-length variations over time.);

9.  determine the mid-March day, using `GISS_srevents(…)` to get the vernal equinox date for calendars in which it varies; and

10. calculate real- and integer-type beginning, middle and ending days using `imon_midbegend(…)` and `rmon_midbegend(…)` for integer- and real-number definitions of the months.

**4.5 Month-length tables and time series**

Tables and time series of month lengths, beginning, middle and ending days, and dates of the vernal equinox can be calculated using the program `month_length.f90`. This program reads an "info file" (`month_length_info.csv`) consisting of an identifying output file name prefix, the calendar type, the beginning and ending simulation age (in years BP), and the age step, and the beginning simulation year (in years CE) and the number of simulation years. Note that in the approach described above, orbital parameters are calculated once per year (step 4 in Sect. 4.4), and are assumed to apply for the whole year. This assumption can lead to small differences (ranging from -0.000863 to 0.000787 days over the past 22 kyr with a mean of -0.00000389 days) in the ending day of one year and the beginning day of the next.

**5 Paleo calendar adjustment**

The objective of the principal calendar-adjustment program `cal_adjust_PMIP3.f90` is to read and clone a "CMIP5/PMIP3"-formatted netCDF file, replacing the original monthly or daily data with calendar-adjusted data, i.e. data aggregated using a fixed-angular calendar appropriate for a particular paleo experiment. In the case of monthly input data, either climatological long-term means or monthly time-series, the data are first interpolated to a daily time step, and then reaggregated to monthly time-step mean values using an appropriate paleo calendar. In the case of daily input data, the interpolation step is obviously





unneeded, and so the data are simply aggregated to the monthly time step. In both cases, new time-coordinate variables are
created (consistent with the paleo calendar), and all other dimension information, coordinate variables and global attributes
are copied, and augmented by other attribute data that indicate that the data have been adjusted. The reading and rewriting of
the netCDF file is handled by subroutines in a module named `CMIP5_netCDF_subs.f90` and various modules and subprograms
for month-length calculations described above are also used here.

**5.1 Interpolation and (re)aggregation**

The pseudo-daily interpolation and (re)aggregation is done using two subroutines `mon_to_day_ts(…)` and `day_to_mon_ts(…)`
in the module `calendar_effects_subs.f90`. The pseudo-daily interpolation is done a year at a time, creating slight
discontinuities between one year and the next in the case of transient or multi-year "snapshot" simulations. The subroutine
`mon_to_day_ts(…)` has options for smoothing those discontinuities, and restoring the long-term mean of the interpolated daily
data to that of the original monthly data.
The (re)aggregation of the daily data is also done a year at a time by collecting the daily data for a particular year, and "padding"
it at the beginning and end with data from the previous and following year if available, as in transient or multi-year simulations
(to accommodate the fact that under some orbital configurations the first day of the current year may occur in the previous
year, or the last day in the following year; Fig. 1). For example, at 6 ka, the changes in the shape of the orbit and the
consequently longer months from January through March (32.5, 29.5 and 32.4 days, respectively) displaces the beginning of
January four days into the previous year, with the last day of December consequently falling just before day 361 in a 365-day
year. In the case of long-term mean "climatological" data ("Aclim" data), the padding is done with ending and beginning days
of the single year of pseudo-daily data.
The calculation of monthly means is done by calculating weighted averages of the days that overlap with a particular month
as defined by the (real-number or fractional) beginning and ending days of that month (from the subroutine
`get_month_lengths(…)`). Each whole day in that interval gets a weight of 1.0, and each partial day gets a weight proportional
to its part of a whole day. It should be noted that in transient simulations, annual averages, constructed either by averaging
actual or pseudo-daily data (or by month-length weighted averages) will differ from the unadjusted data.

**5.2 Processing individual netCDF files**

At present, the program reads an "info file" that provides file and variable details, and this info file will be easily modified to
accommodate "CMIP6/PMIP4" formatted files (https://pcmdi.llnl.gov/CMIP6/Guide/modelers.html#5-model-output-
requirements) as they become available. The fields in the info file include (for each netCDF file), the variable (e.g. "tas",
"pr"), the "realm-plus-time-frequency" type (e.g. "Amon", "Aclim", …), the model name, the experiment name (e.g.
"midHolocene"), the ensemble member (e.g. "r1i1p1"), and the simulation year beginning date and ending date (as a

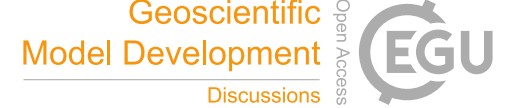



YYYYMM or YYYYMMDD string).  An input filename "suffix" field is also read (which is usually blank, but is "-clim" for
Aclim-type files), as is an output filename "suffix" field (e.g. "_cal_adj"), which is added to the output filename to indicate
that it has been modified from the original.  The info file also contains the simulation age beginning and end (in years BP), the
increment between simulation ages (usually 1 in the application here), the beginning simulation year (years CE) and the number
of simulation years. This information could also be gotten by parsing the netCDF file name and reading the calendar attribute
and time-coordinate variables, but that would add to the complexity of the program.
The output netCDF files have the string " _cal_adj" appended to the end of the filename.  In the case of monthly time series
(e.g. "Amon") or long-term means (e.g. "Aclim") the file names are otherwise the same as the input data.  In the case of the
daily input data, with "day" as the "realm plus time frequency" string, that string is changed to "Amon2".
The adjustment of a file using `cal_adjust_PMIP3.f90` includes the following steps:

1.  read the info file, construct various file names, allocate month-length variables;
2.  generate month lengths using the subroutine `get_month_lengths(….)`;
3.  open input and output netCDF files; and for each file
4.  redefine the time-coordinate variable as appropriate using the subroutines `new_time_day(…)` and `new_time_month(…)`

in the module `CMIP5_netCDF_subs.f90`;

5.  create the new netCDF file, copy the dimension and global attributes from the input file using the subroutine

`copy_dims_and_glatts(…)`, define the output variable using the subroutine `define_outvar(…)`;

6.  get the input variable to be adjusted;
7.  for each model grid point, get calendar-adjusted values as described above using the subroutines `mon_to_day_ts(…)`

and `day_to_mon_ts(…)`; and

8.  write out the adjusted data, and close the output file.
**5.3 Further examples**
Five other main programs that serve as "drivers" for some of the subroutines or that demonstrate particular aspects of
procedures used here are included in the GitHub repository for the programs (https://github.com/pjbartlein/PaleoCalAdjust):
▪  `GISS_orbpar_driver.f90`  and  `GISS_srevents_driver.f90`;  Main  programs  that  call  the  subroutines

`GISS_orbpars(…)` and `GISS_srevents(…)` to produce tables of orbital parameters and "solar events" like the dates of

equinoxes, solstices and perihelion and aphelion.

▪  `demo_01_pseudo_daily_interp.f90`; Main program that demonstrates linear and mean-preserving pseudo-daily

interpolation.

▪  `demo_02_adjust_1yr.f90`; Main program that demonstrates the paleo calendar adjustment of a single year's data.

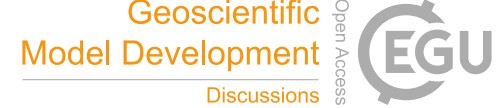

▪    `demo_03_adjust_TraCE_ts.f90`; Main program that demonstrates the adjustment of a 22040 year-long time series of
monthly TraCE-21k data.

## 6 Summary

As has been done previously (e.g. Kutzbach and Otto-Bliesner, 1982; Kutzbach and Gallimore, 1988; Jousaumme and
Braconnot, 1997; Pollard and Reusch, 2002; Timm et al., 2008; Chen et al., 2011; Kageyama et al., 2018), we have described
the substantial impacts of the paleo calendar effect on the analysis of climate-model simulations, and provide what we hope is
a straightforward way of making adjustments that incorporate the effect. The interval between previous calls to include
consideration of the calendar effect in paleoclimate analyses has ranged between three and nine years over the past nearly four
decades, with a median interval of six years. The size and impact of the calendar effect warrant its consideration in the analysis
of paleo simulations, and we hope that by providing a relatively easy-to-implement method, that will become the case.

**Code and data availability**

The Fortran 90 source code (main programs and modules), example data sets, and the data used to construct the figures are
available from Zenodo (https://zenodo.org/) at the following URL: https://doi.org/10.5281/zenodo.1478824 and .from GitHub
(https://github.com/pjbartlein/PaleoCalAdjust). All climate data used here are available for download at the URLs cited in the
text.

**Author contribution**

PB designed the study, developed the Fortran 90 programs, and wrote the first draft of the manuscript. Both authors contributed
to the final version of the text.

**Competing Interests**

The authors declare that they have no conflict of interest.

**Acknowledgements:** We thank Jay Alder, Martin Claussen, and Anne Dallmeyer for their comments on earlier versions of
the text. This publication is a contribution to PMIP4. TraCE-21ka was made possible by the DOE INCITE computing
program, and supported by NCAR, the NSF P2C2 program, and the DOE Abrupt Change and EaSM programs. CMAP
precipitation data were provided by the NOAA/OAR/ESRL PSD, Boulder, Colorado, USA, from their Web site at
https://www.esrl.noaa.gov/psd/.      CFSR    near-surface    air-temperature    data    were    obtained    from
https://esgf.nccs.nasa.gov/projects/ana4mips/ (for the original source see http://cfs.ncep.noaa.gov). Maps were prepared using

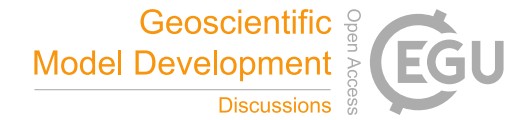



NCL, the NCAR Command Language (Version 6.4.0 [Software], 2017, Boulder, Colorado: UCAR/NCAR/CISL/TDD.
http://dx.doi.org/10.5065/D6WD3XH5). S.S. was supported by the U.S. Geological Survey Land Change Science Program.
Any use of trade, firm, or product names is for descriptive purposes only and does not imply endorsement by the U.S.
Government.

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






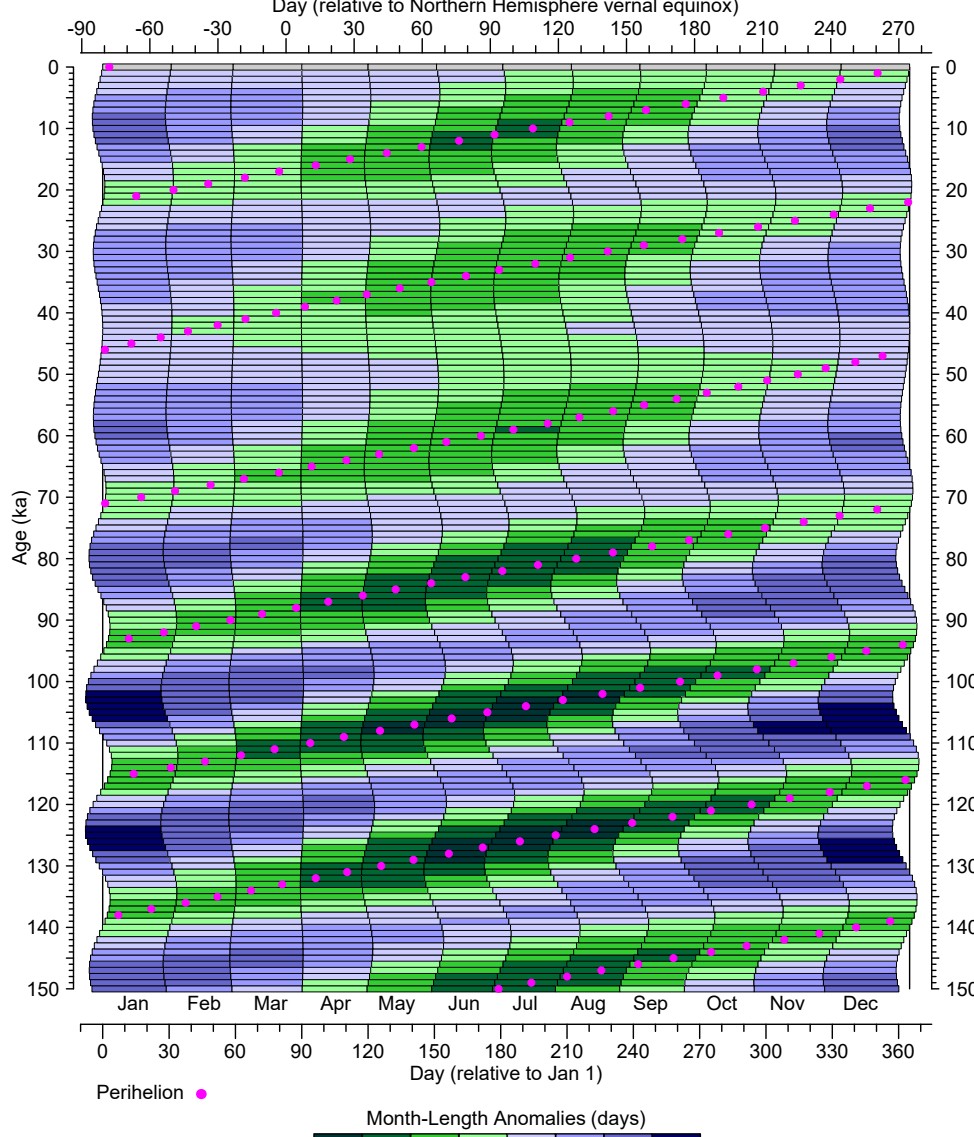


**Figure 1.** Variations over the past 150 kyr in the beginning and ending days of fixed-angular months for a 365-day "noleap" calendar, shown for 1 kyr intervals beginning at 0 ka (1950 CE). The left side of each horizontal bar shows the beginning day while the right side shows the ending day of a particular month for each 1 kyr interval. The month-length "anomalies" or differences from the present-day are shown by shading, with individual paleo months that are shorter than those at present indicated by green shades and those that are longer indicated by blue shades. The day that perihelion occurs for each 1 kyr interval is indicated by a magenta dot, and the overall pattern of month-length anomalies can be seen to follow the day of perihelion. The figure shows that the changing month lengths move the beginning, middle and ending days of each month (as well as the beginning and ending days of the year).





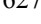

627

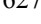

628

**Figure 2.** Variations in the difference (in days) between the mid-month day of each month and the day of the June solstice. Months that are shifted closer to the June solstice are indicated by orange hues while those that are farther away are indicated by blue. As in Fig. 1, variations over the past 150 kyr in the beginning and ending days of fixed-angular months for a 365-day "noleap" calendar are shown for 1 kyr intervals beginning at 0 ka (1950 CE). The left side of each horizontal bar shows the beginning day while the right side shows the ending day of a particular month for each 1 kyr interval. Variations in the beginning and ending days of individual months can be seen to track the climatic precession parameter ($e \cdot sin\ \omega$, where e is eccentricity and $\omega$ is the longitude of perihelion measured from the vernal equinox, an index of Earth's distance from the Sun at the summer solstice), which is plotted at the right side of the figure (red dots). (Note that the inverse of the climatic precession parameter is plotted for easier comparison.)



637

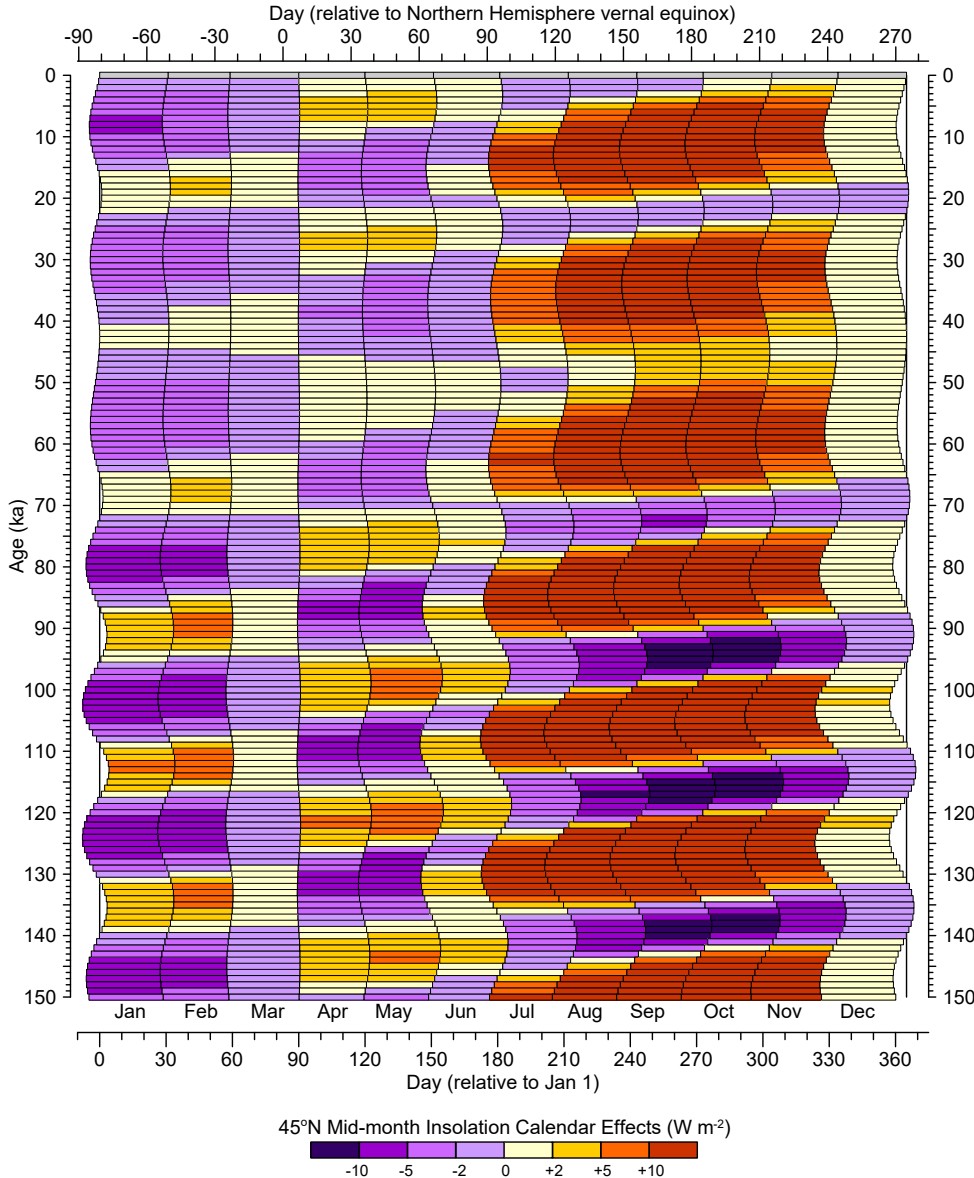

638

**Figure 3.** Calendar effects on insolation at 45° N. The differences plotted show the values of average daily insolation at mid-month days identified using the appropriate fixed-angular paleo calendar minus those using the fixed-length definition of present-day months, with orange hues showing positive difference, and purple hues negative. As in Fig. 1, variations over the past 150 kyr in the beginning and ending days of fixed-angular months for a 365-day "noleap" calendar are shown for 1 kyr intervals beginning at 0 ka (1950 CE). The left side of each horizontal bar shows the beginning day while the right side shows the ending day of a particular month for each 1 kyr interval.








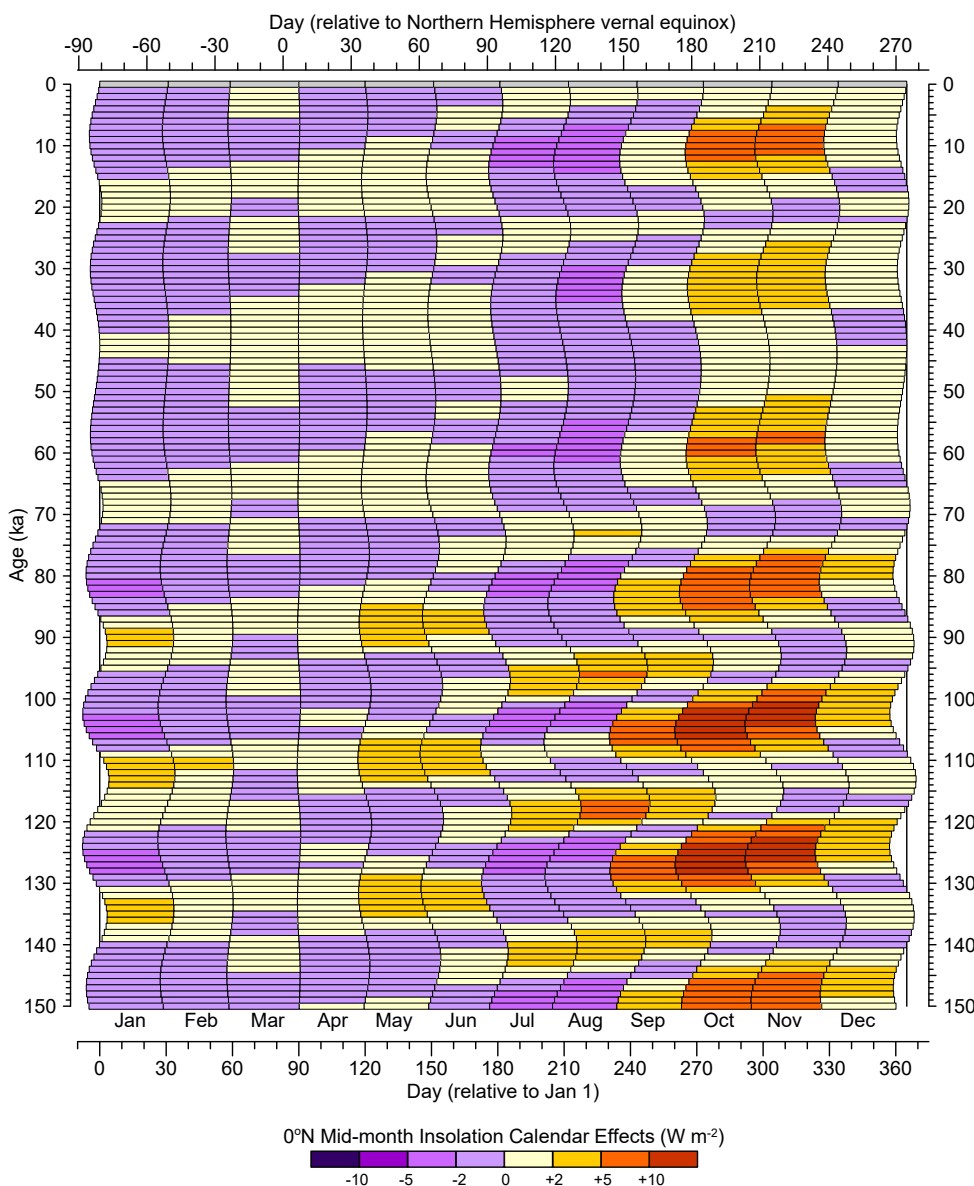


**Figure 4.** Calendar effects on insolation at the equator. The differences plotted show the values of average daily insolation at mid-month days identified using the appropriate fixed-angular paleo calendar minus those using the fixed-length definition of present-day months, with orange hues showing positive difference, and purple hues negative. As in Fig. 1, variations over the past 150 kyr in the beginning and ending days of fixed-angular months for a 365-day "noleap" calendar are shown for 1 kyr intervals beginning at 0 ka (1950 CE). The left side of each horizontal bar shows the beginning day while the right side shows the ending day of a particular month for each 1 kyr interval.



653

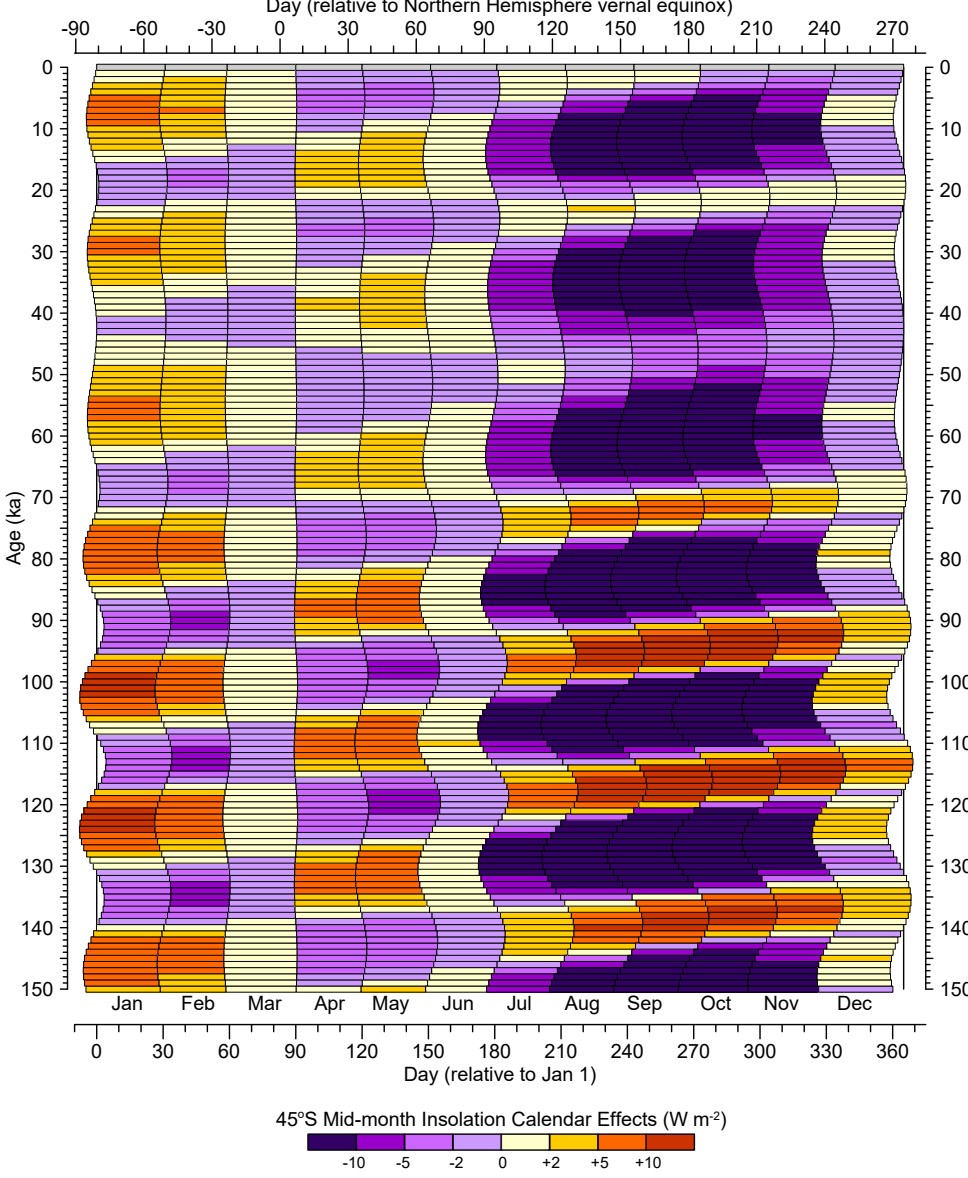

654

**Figure 5.** Calendar effects on insolation at 45° S. The differences plotted show the values of average daily insolation at mid-month days identified using the appropriate fixed-angular paleo calendar minus those using the fixed-length definition of present-day months, with orange hues showing positive difference, and purple hues negative difference. As in Fig. 1, variations over the past 150 kyr in the beginning and ending days of fixed-angular months for a 365-day "noleap" calendar are shown for 1 kyr intervals beginning at 0 ka (1950 CE). The left side of each horizontal bar shows the beginning day while the right side shows the ending day of a particular month for each 1 kyr interval.








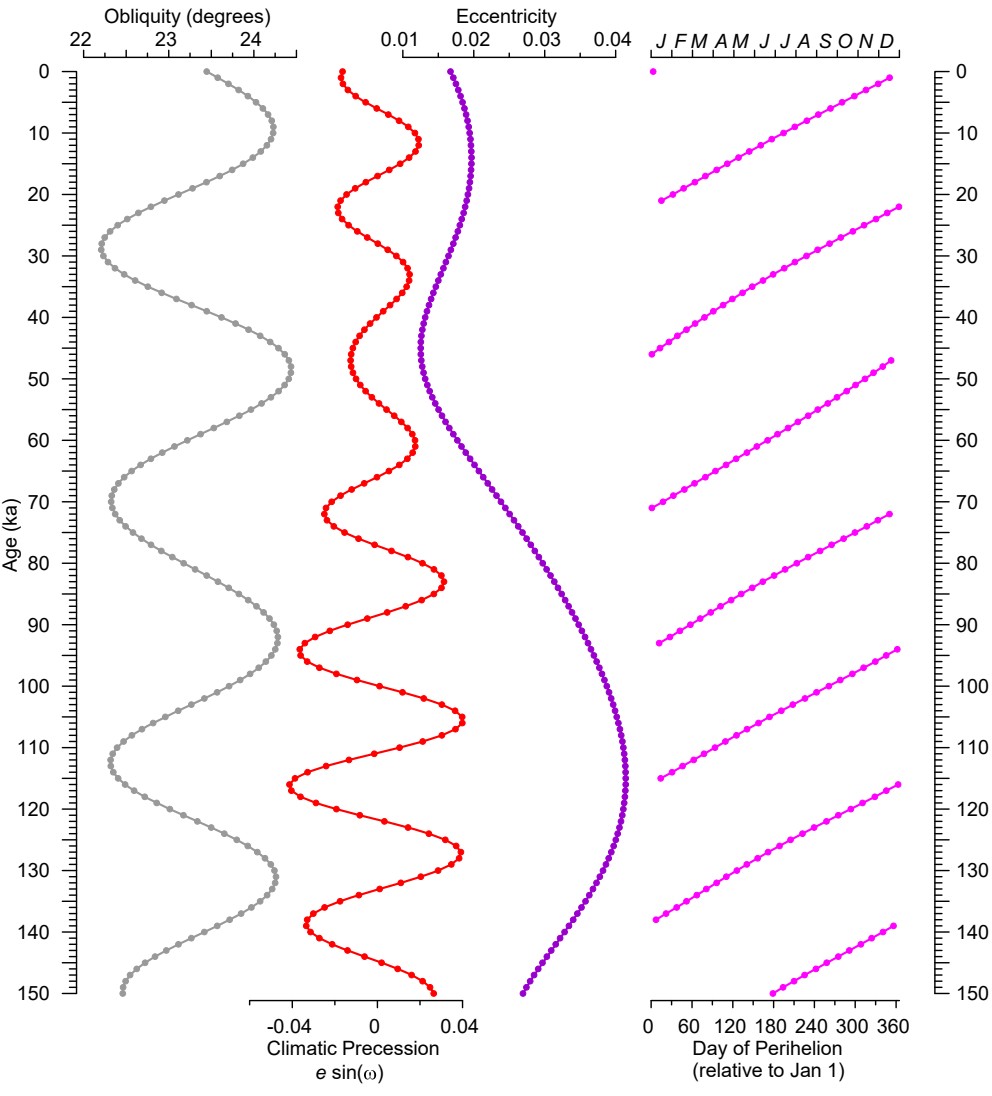


**Figure 6.** Orbital parameter variations at 1 kyr intervals over the past 150 kyr for obliquity, climatic precession, eccentricity, and day of perihelion (relative to January 1). Climatic precession is calculated as e · sin(ω), where e is eccentricity and ω is the longitude of perihelion measured from the vernal equinox.







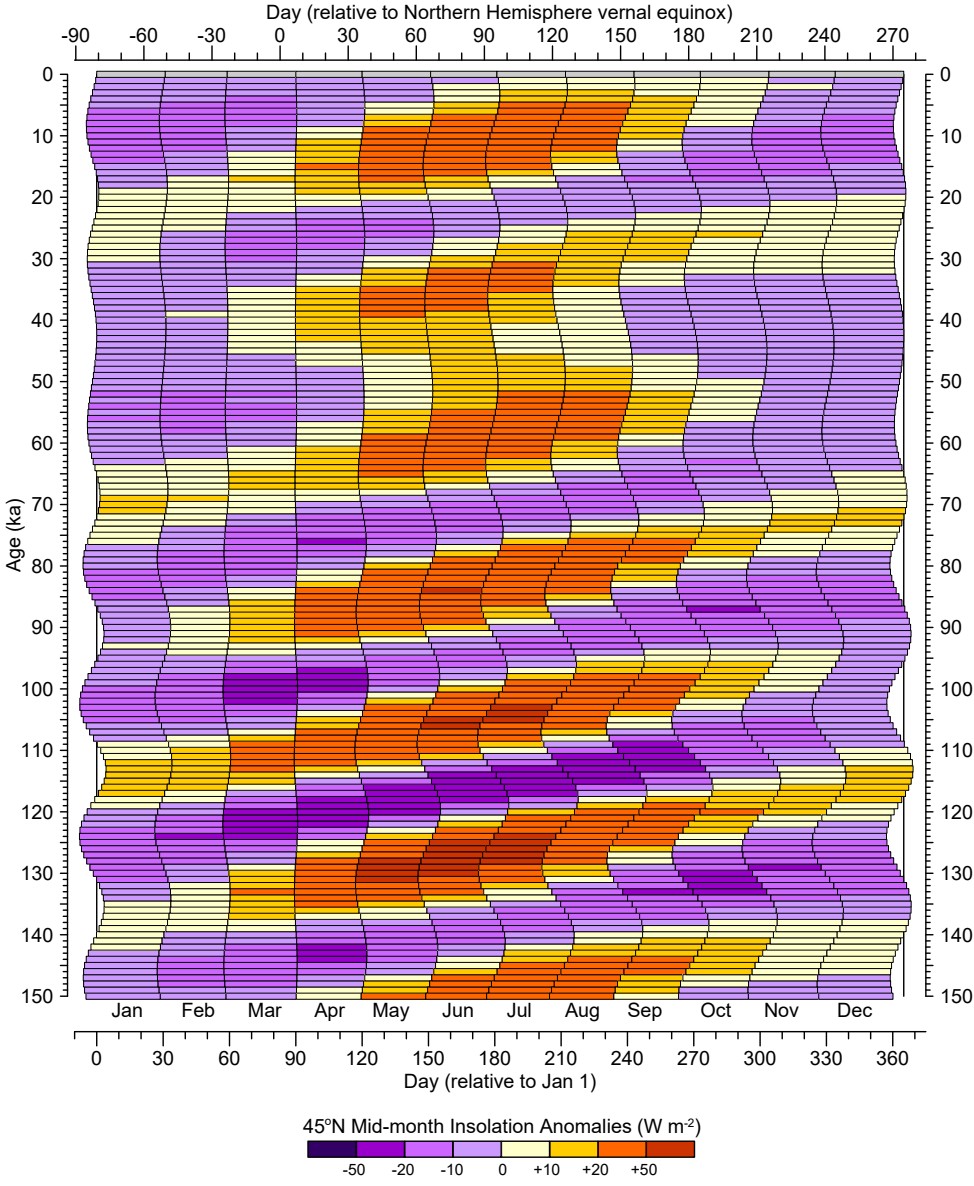

**Figure 7.** Long-term differences in mid-month average daily insolation relative to present (0 ka or 1950 CE) at 45° N for a fixed-angular calendar. As in Fig. 1, variations over the past 150 kyr in the beginning and ending days of fixed-angular months for a 365-day "noleap" calendar are shown for 1 kyr intervals beginning at 0 ka (1950 CE). The left side of each horizontal bar shows the beginning day while the right side shows the ending day of a particular month for each 1 kyr interval.







**Figure 8.** Long-term differences in mid-month average daily insolation relative to present (0 ka or 1950 CE) at the equator for a fixed-angular calendar. As in Fig. 1, variations over the past 150 kyr in the beginning and ending days of fixed-angular months for a 365-day "noleap" calendar are shown for 1 kyr intervals beginning at 0 ka (1950 CE). The left side of each horizontal bar shows the beginning day while the right side shows the ending day of a particular month for each 1 kyr interval.



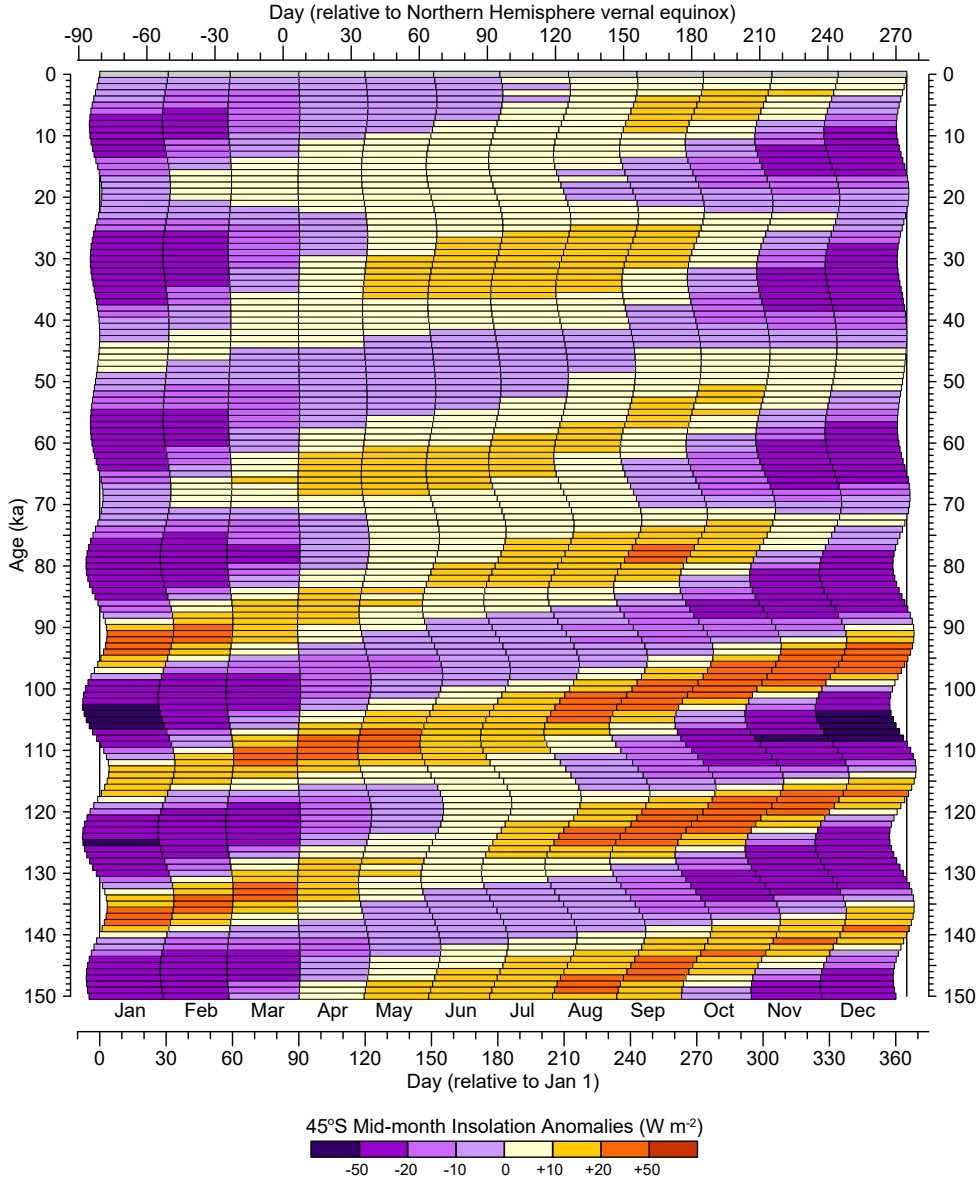

**Figure 9.** Long-term differences in mid-month average daily insolation relative to present (0 ka or 1950 CE) at 45° S for a fixed-angular calendar. As in Fig. 1, variations over the past 150 kyr in the beginning and ending days of fixed-angular months for a 365-day "noleap" calendar are shown for 1 kyr intervals beginning at 0 ka (1950 CE). The left side of each horizontal bar shows the beginning day while the right side shows the ending day of a particular month for each 1 kyr interval.





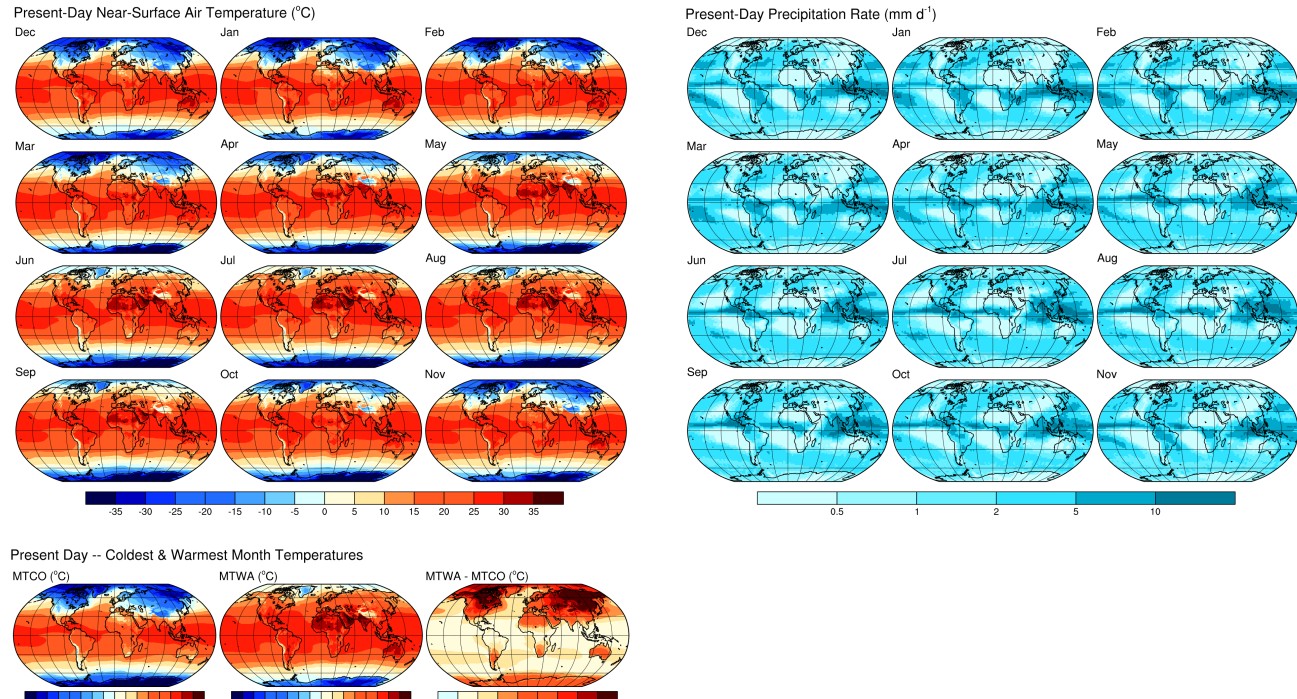


**Figure 10.** Present-day (1981-2010 CE) long-term mean values of monthly near-surface air temperature (*tas*) from the Climate Forecast
System Reanalysis (CFSR), the mean temperatures of the warmest and coldest months and their differences from the same data, and
precipitation rate (*precip*) from the CPC Merged Analysis of Precipitation (CMAP).











**Figure 11.** Calendar effects on near-surface air temperature for 6 ka (upper left), 97 ka (upper right), 127 ka (lower left) and 116 ka (lower right). The maps show the patterns of month-length adjusted average temperatures minus the unadjusted values, using 1981-2010 long-term averages of CFSR *tas* values, with positive difference (indicating that the adjusted data would be warmer than unadjusted data) in red hues, and negative differences in blue.

704


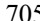

705

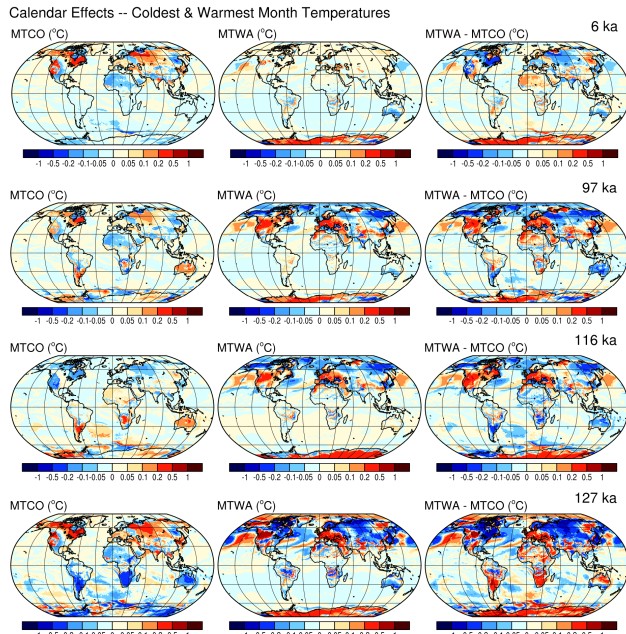

706

**Figure 12.** Calendar effects on the mean near-surface air temperatures of the warmest (MTWA) and coldest (MTCO) months and their differences (an index of seasonality) for 6 ka, 97 ka, 116 ka and 127 ka (top to bottom row). The maps show the patterns of month-length adjusted average temperatures minus the unadjusted values for MTWA and MTCO, using 1981-2010 long-term averages of CFSR *tas* values, with positive difference (indicating that the adjusted data would be warmer than unadjusted data) in red hues, and negative differences in blue.






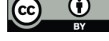



**Figure 13.** Calendar effects on precipitation rate for 6 ka (upper left), 97 ka (upper right), 127 ka (lower left) and 116 ka (lower right). The maps show the patterns of month-length adjusted precipitation rate minus the unadjusted values, using 1981-2010 long-term averages of CMAP *precip* values, with positive difference (indicating that the adjusted data would be wetter than unadjusted data) in blue hues, and negative differences in brown.

720







**Figure 14.** Time series of original and month-length-adjusted annual area-weighted averages of TraCE-21k data (Liu et al., 2009), expressed as difference from the 1961-1989 long-term mean for (a-c) 2 m air temperature, (d) precipitation rate, and (e-f) precipitation minus evaporation (P - E). The original or unadjusted data are plotted in gray and black, and the adjusted data in colors. The area averages are grid-cell area-weighted values for land grid points in each region, and the smoother curves are locally weighted regression curves with a window half-width of 100 years. The regions are defined as: (a) 15 to 75° N and -170 to 60° E, (b) 10 to 50° S and 110 to 160° E, (c) global ice-free land area, (d) 0 to 30° N and -30 to 120° E, (e) 5 to 17° N and -5 to 30° E, and (f) 31 to 43° N and -5 to 30° E.

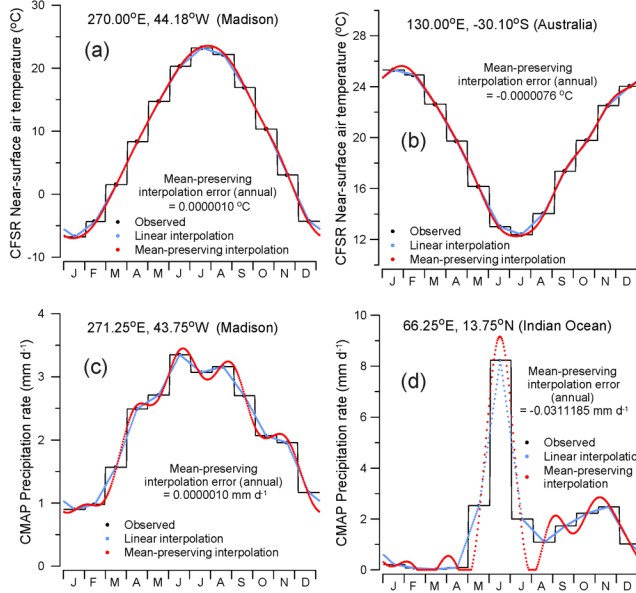

**Figure 15.** Pseudo-daily interpolated temperature (top row) and precipitation (bottom row) for some representative locations: (**a**, **c**) Madison, Wisconsin, USA, (**b**) Australia, and (**d**) the Indian Ocean. The original monthly mean data are shown by the black dots and stepped curves (black lines), daily values linearly interpolated between the monthly mean values are shown in blue, and daily values using the mean-preserving approach of Epstein (1991) are shown in red. The annual interpolation error (or the difference between the annual average calculated using the original data and the pseudo-daily interpolated data) is given for the mean-preserving approach in each case. The interpolated data for this figure were generated using the program `demo_01_pseudo_daily_interp.f90`.





**Figure 16.** Pseudo-daily interpolation errors for CFSR near-surface air temperature (left-hand column) and CMAP precipitation rate (right-hand column). The top set of maps shows the interpolation errors, or the differences between the original monthly mean values and the monthly mean values recalculated from linear interpolation of pseudo-daily values. The bottom set of maps shows the interpolation errors for mean-preserving (Epstein, 1991) interpolation.