# Peer review of "Paleo calendar-effect adjustments in time-slice and transient climate-model simulations (PaleoCalAdjust v1.0): impact and strategies for data analysis"

_Geoscientific Model Development, 2018_

## Referee Comment (RC1) · Anonymous Referee #1 · 4 Jan 2019

**1 Summary Statement**

This paper is a valuable contribution both in terms of raising awareness for the calendar definition problem in simulations with different orbital configurations and providing computer programs (plus source code) for easy conversions of climate model output data from the traditional fixed calendar system to angular-aligned equivalent monthly mean values. The support of NetCDF CF-conform model data will enable climate modelers and users of paleoclimate model data to provide or calculate on their own these

angular-based monthly mean climate data.

The examples and figures shown in this article serve the purpose of illustrating the cause of the problem. More so, the authors apply the angular-based calendar definition to present-day climate data to highlight the pure bias resulting from the shift in the seasons when orbital changes in precession (plus eccentricity) shift the longitude of the perihelion.

The article is well-written, and most figures are easy to comprehend. A few exceptions, where improvements should be made in the text /figure, I would like to point out.

Overall, I have only minor comments and a few critical points to raise in the following section.

**2 Specific comments**

Line 30-32: I believe you meant to express just the opposite relationship here. Closer to the sun, the Earth covers a wider angle in the same amount of time (Kepler's 2nd law), and therefore a 90-deg angle will be passed in a shorter time when Earth is near the Perihelion point.

Line 36-40: A citation to previous literature on this topic should be added.

Line 43-46: Could you cite some examples from the literature, or else point out that you'll demonstrate this here in the later sections.

Line 57-68: I appreciate that you mention the other methods, too. It would have been nice to see differences between your method and the method of Chen et al (2011) later in the text, too. Their code (and I believe Pollard and Reusch (2002), too) could be tested on the reanalysis data set, too.

Section 2: You mention the 360-day and 365-day calendar years, it would be worth

pointing out that the default code and the shown results are all (?) based on 365-day calendar. Or was it with the actual leap-year calendar, since you worked with reanalysis data.

Line 119 and Fig.3: The color scale in the Figure is truncated at 10 Wm$^{-2}$, but the text describes values up to 35 Wm$^{-2}$. Please consider an update of the color scale range in the Figure. This may also apply for other figures. Please revisit the color scales and make sure that the range covers the actual value range of the data.

Line 127: Here I feel that it is possible to give a more precise value than 10Wm$^{-2}$, given the range of the color scales in the Figures 7-9.

Lines 254-269: The description of the effects of the biases in the transient simulation is done well. However, in the discussion section (or here in this paragraph) the authors should discuss the implications for real-world paleoclimate studies. Two things come into my mind: (a) most often paleoclimate studies use seasonal mean data rather than single month data, (b) for proxy model comparisons and interpretation of time series signals one could as: 'Does it matter in the end?' Most likely one does encounter seasonal rather than single month responses in proxies, and processes within the climate system are often analyzed (at least at first) by season rather than single months. I still find it appropriate to present the monthly mean results here, but the practical points of view should also be highlighted.

Section 4:

I got a bit confused with the technical definition of start day of a year. What is actually determining the first day of a calendar year? Astronomically, vernal equinox for example would make sense, but here I believe it is defined more on technical grounds by the default application and historical developments in climate models? That is, in the present-day calendar we call Jan 1st the first day in a year, which is a certain longitude position of the Earth in its orbit around the sun. Please mention or explain the basis of the definition start day.

Line 369: the equation is for a variable with physical units of time. On the right-hand side, is the variable phi (an angle in degrees) correct?

Line 373: "fixed at 80 days after the beginning of the model year"

I am not sure if that is clear to all readers, it got me thinking again, what is actually better: To describe the begin of the year as fixed relative to the vernal equinox point, and defined via a longitude angle; or from a modeler's perspective where we are used to think of years starting Jan 1st, and vernal equinox is somehow flexible and can be set to a specific day in the year.

Line 373: better if you mention here the actual sub-routine that is to be used with the 365-day calendars.

Section 5: Can you briefly mention which libraries and versions (I think of udunits, netcdf) you used and recommend in connection with the code?

**3  Figures**

**Figure 2:** Can you add the summer solstice as a point in the figure? It could be helpful to have in this figure. One question I have though, since there is a little problem:

When you compare Dec and Jan the color code switches from plus (blue) to minus (red) colors. At a day in the year 180 days before (after) summer solstice (SS) [in a 360-day calendar] it becomes meaningless to talk about closer and farther relative to SS. An anomaly of one day brings that day closer to SS and at the same time farther away from SS. Can you please explain once more how the reader should interpret the color code in Figure 2? And maybe explain if that has implications on the definition of anomalies shown in Figure 3-5, or not?

**Figure 3 (and following Fig 4,5):**

Why is it that all but the Dec months show consistent color coding with Figure 2? In Fig. 3 the December months clearly stand out because they do not vary in terms of anomalies over the past 150,000 years. I also find it inconsistent with the maps in Figure 11.
* * *

---

## Referee Comment (RC2) · Anonymous Referee #2 · 8 Jan 2019

This manuscript discusses the problem associated with the usual definition of seasons in climate models, when simulating climates under different orbital (precessional) configurations, a situation rather common in paleoclimatology. The subject is important, sometimes even critical, and too often it is overlooked. So this discussion and more importantly the availability of some useful routines is a significant and valuable contribution. Unfortunately, I have serious critical comments, both on the manuscript itself, but also on the methodology used in the mathematical routines. With these shortcomings I cannot recommend publication of the manuscript in its current state.

Major comments

1 – Solving the astronomical problem (§4.2 – lines 350 to 377) The question of the length of seasons is a very classical astronomical problem, since at least the antiquity. It was solved, in the two-body approximation, by Kepler (1609) with the famous first and second laws. The mathematical problem is called "Kepler's equation": $M = E - e \sin E$ where M is the mean anomaly (basically time), e is the eccentricity and E is the eccentric anomaly (basically the orbital position). The direct problem (ie. computing M or time for given orbital positions E like equinoxes or solstice for the seasons) is trivial. Solving the inverse problem (computing position E as a function of time M) is likely to be part of the climate model code (usually in a routine called "solar". . . but see below), and numerous algorithms have been proposed during the last four centuries to solve it exactly. In this context, it is extremely disturbing to see a scientific paper based on a very crude approximation of Kepler's equation (manuscript equations 1 and 2) based on a first order expansion in e. To be more explicit (see standard textbooks. . .): $M = (2\pi/T) t$ $E = 2 \text{Arctan}[ \sqrt{((1-e)/(1+e))} \tan(v/2) ]$ $v = \lambda - \ddot{I}\acute{U} + \pi$ (T = 1 year, t = time from perihelion, v = angular orbital position from perihelion, $\lambda$ = angular orbital position from some reference, usually spring equinox, $\ddot{I}\acute{U}$ = climatic precession, or angular orbital position of perihelion versus a reference, usually vernal point, $\pi = 180°$ = spring equinox vs. vernal point) So, for given orbital parameters (precession $\ddot{I}\acute{U}$ ; eccentricity e) it is rather simple to compute time t as a function of $\lambda$ using Kepler's equation. I do not understand the need for a first order approximation in e when equipped with a computer. . . Where does it come from ? Kutzbach and Gallimore (1988); citing Symon (1964). . . probably citing someone who had no pocket calculator and found it useful to write 1st order approximations. This is no more relevant and, I think, not acceptable today : Arctan, tan, Sqrt are usually available in most computer languages. . .

2 – The same kind of unacceptable "pedestrian" procedure is used (line 374) for solving the equation sin(x)=-1 Using standard mathematics, the solution is x = -$\pi$/2 The corresponding algorithm described in the manuscript appears, to say the least, a bit

awkward: Âń Select $\varphi$p that minimizes {-1-sin(($2\pi/360$)($\varphi$ - $\varphi$p))} Âż I prefer to write it more simply as Âń $\varphi$p = $\varphi$ + $\pi/2$ (or in degrees $\varphi$p = $\varphi$ + $90°$) Âż and would not use a computer minimization routine for that. . .

Overall, I do not understand why the whole procedure is so complex, and divided in five computer procedures (Step 1 to 5). It is based on unnecessary approximations and on very awkward procedures like ". . .advancing along the orbit at 0.001-day increments. . .". As a result, I have some severe reservations on the relevance or validity of the code. As explained above, the time between 2 orbital positions is just a simple application of Kepler's equation: t2 - t1 =(T/2$\pi$)(M2 - M1) = (T/2$\pi$)( E2 − e sinE2 − E1 + e sinE1 ) = ... with (E1, E2) functions of orbital positions ($\lambda$1, $\lambda$2) as explained above. This should stand in 2 or 3 lines of computer code, no more.

3 − A large part of the manuscript is about describing the "impact of the calendar effect" (§3 − lines 133 to 300). Most of this has been already described and discussed in numerous papers. I believe this part is far too long and far too descriptive. Again, most of this is known since quite a long time: geologists in the XIXth and early XXth centuries were discussing the astronomical forcing not in radiative terms (W/m2) but in terms of season duration (number of days of winter versus summer). This "calendar effect" is in fact at the heart of the precessional forcing. The key point is not so much its existence or its relevance, but how to deal with it in order to interpret model results in seasonal terms.

4 - Lines 320-348. I do not understand why the "mean-preserving" interpolation does not preserve exactly the mean. The fact that the error is small (but can be as big as -0.12 mm/day) is not reassuring. Where does the error come from? If there is no other mathematical treatment that the "mean-preserving interpolation", then I expect the mean to be exactly preserved, not approximately.

Minor comments

line 30-31 : Âń larger number of days Âż. . .. Should be the opposite.

Please state somewhere in the introduction that a fixed angular month is precisely 30°. Stating that a "fixed-angular month corresponds to a fixed number of degrees" is not sufficient. It should also be stated that all "fixed-angular month" are equal, and therefore equivalent to 360°/12. Without such a clear statement, the definition of a "fixed-angular month" is very ambiguous.

line 360 : Âń To calculate the orbital parameters . . . we adapted a set of programs . . . https://data.giss.nasa.gov/ar5/solar.html Âż This link is broken. (I am pretty sure this solves Kepler's equation. . .)

line 369 : equation (2) there is a missing index p in the second cosine (see the original ref).
* * *

---

## Referee Comment (RC3) · Anonymous Referee #3 · 17 Jan 2019

The authors present the software which can convert the standard output of PMIP models to the new, celestial, calendar. The authors believe that such a calendar is better than the standard (fixed day) one. This is my main disagreement with the authors and other reviewers. I simply do not believe that the celestial calendar is better (or worse) than the standard one.

The authors begin their paper from the statement that "there are two ways of defining month or seasons" (p. 1). This is of course not true since there is a myriad of ways to define months and seasons. Julian and Gregorian calendars are obvious examples.

[Figure]

For paleoclimate applications, there are many other options. For example, one can set the summer solstice to 22 June instead of setting vernal equinox to March 21, as is required by the PMIP protocols. In fact, fixing of the summer solstice would be more reasonable, at least for the Northern Hemisphere. While for the present day the calendar has absolute meaning since observational climate data used for model validations are aggregated according to the "official" months, for the analysis of past climate simulations in principle one is free which calendar to use. For model intercomparison, the only important requirement is that all models should use the same calendar. For comparison with paleoclimate proxies, any calendar is of limited use because the calendar is human invention and Nature has no idea about seasons or months. Therefore model/data comparison cannot be improved by choosing the "right" calendar. To the contrary, proper model/data comparison requires abandoning of any calendar and using climate characteristics which are independent of the choice of the calendar. Of course, ideally observed proxy records should be compared with the simulated ones.

Let's consider the advantages of using celestial calendar compare to the standard (fixed-day) one. For two special orbital configurations, namely, when summer solstice coincides with perihelion or aphelian ("warm" and "cold" orbits respectively) celestial calendar has one obvious advantage –the maxima and minima of insolation will always occur at the same days (90 and 270 days of celestial calendar) while under large eccentricity when using the standard calendar, the summer solstice (and maxima/minima of insolation) can deviate from 22 June by +-5 days. However, for the two "representative" months, January and July, the differences between the standard and celestial calendars (as shown by numerous figures in the Bartlein and Shafer manuscript) are rather small. These differences increase significantly during the transition months (August-November). Which of two calendars is better for these months? The simple answer is NONE because these months exist only in our imagination and I cannot see any sense in comparison, for example, September temperatures at present and 127 000 years ago. However, other workers may disagree with me and want to analyze climate change during spring or autumn. In this case, they have to realize that for these

months, the celestial calendar has a serious problem even compare to the standard (fixed-day) one because it corrupts the most fundamental characteristics of the real world – time. For a high eccentricity, the days in the celestial calendar can be 10% shorter or longer than the real ones and, as the result, the beginning for example of celestial "October" can move back and forward compare to the summer solstice by more than 10 "real" days (Fig.2). At the same time, the internal time scales of the climate system do not depend on the orbital parameters and therefore the time lags between insolation and climate characteristics remain nearly constant in the real time, not in the celestial "days". Thus using of celestial calendar corrupts the physics of climate. It is noteworthy that in the paper by Kutzbach and Gallimore (1988) cited in the manuscript and where celestial calendar has been used, Kutzbach and Gallimore explicitly stated (page 820, first para) about the celestial calendar:

"The procedure, however, is mainly applicable to climate experiments that prescribe ocean and sea ice conditions, i.e., climate systems not having interactive components with significantly different lags in response to solar forcing".

Thus Kutzbach and Gallimore already 30 years ago clearly realized that corruption of absolute time is a serious problem. Surprisingly, modern authors seem to be unaware of this problem.

Above I only discussed the situation with two very specific orbital configurations –when summer equinox occurs in perihelion or aphelion (like that at 126 ka or 116 ka). What about an arbitrary Earth's orbit? For any arbitrary orbit, the only advantage of the celestial calendar disappears because maxima and minima of insolation at different latitudes do not coincide anymore with the solstices and can deviate from them by up to one week, i.e. as much as they can deviate from 22 June and 22 December in the standard calendar.

By saying that, I want to make it clear that I am not against using several different calendars. This at least helps to understand that at the orbital time scales, things

like "spring" or "October" do not have any meaning. But to be useful, the manuscript under consideration should not make false impression that it presents The Solution for the Calendar Problem and that Celestial Calendar is the right one. I believe, the manuscript requires a thorough discussion of problems and limitations of any calendar applied to the analysis of model results.

———————————————————

---

## Author Comment (AC1) · 27 Feb 2019

Author Comment 1

We thank the referees for their thoughtful comments and respond to their individual comments below. The modifications we have made to the code and will be making to the manuscript are summarized here:

1) After the paper was submitted, we modified the "info-file" format and input/output of the program `cal_adjust_PMIP3.f90` to accommodate CMIP6/PMIP4-type files, which have an additional field in their filenames specifying the model's grid. We also changed the program file name from `cal_adjust_PMIP3.f90` to `cal_adjust_PMIP.f90` so that the name of the program now reflects its ability to handle both PMIP3- and PMIP4-type files. The current version of the code is available in the GitHub code repository:

   (https://github.com/pjbartlein/PaleoCalAdjust).

2) We adopted Referee 2's suggestion of using Kepler's equation directly, as opposed to the approximation used in Kutzbach and Gallimore (1988). Doing so produced no practical difference in the results. A reader may compare, for example, the figures from any earlier release of the code and data at GitHub or Zenodo (https://doi.org/10.5281/zenodo.1478824; v1.0, v1.0a, and v1.0b) with the current release (v1.0c) to verify this assertion.

3) Based on comments of other referees and users, we have also edited the code to improve its transparency and organization. These changes can also be noted by comparison of the different code releases.

In the following text, the referees' comments are in italic font followed by our authors' responses in regular font.

*Anonymous Referee #1*

*Summary Statement*

*This paper is a valuable contribution both in terms of raising awareness for the calendar definition problem in simulations with different orbital configurations and providing computer programs (plus source code) for easy conversions of climate model output data from the traditional fixed calendar system to angular-aligned equivalent monthly mean values. The support of NetCDF CF-conform model data will enable climate modelers and users of paleoclimate model data to provide or calculate on their own these angular-based monthly mean climate data.*

*The examples and figures shown in this article serve the purpose of illustrating the cause of the problem. More so, the authors apply the angular-based calendar definition to present-day climate data to highlight the pure bias resulting from the shift in the seasons when orbital changes in precession (plus eccentricity) shift the longitude of the perihelion.*

*The article is well-written, and most figures are easy to comprehend. A few exceptions, where improvements should be made in the text /figure, I would like to point out.*

*Overall, I have only minor comments and a few critical points to raise in the following section.*

*Specific comments*

*Line 30-32: I believe you meant to express just the opposite relationship here. Closer to the sun, the Earth covers a wider angle in the same amount of time (Kepler's 2nd law), and therefore a 90-deg angle will be passed in a shorter time when Earth is near the Perihelion point.*

Yes. We will change the wording to avoid ambiguity.

*Line 36-40: A citation to previous literature on this topic should be added.*

We will add citations to the previous literature (e.g. Kutzbach and Gallimore, 1988; Jousaume and Braconnot, 1997).

*Line 43-46: Could you cite some examples from the literature, or else point out that you'll demonstrate this here in the later sections.*

To keep the introduction focused on the main issue, we will add a forward reference to Sections 3.1 to 3.3, where we describe the map patterns of the calendar effect.

*Line 57-68: I appreciate that you mention the other methods, too. It would have been nice to see differences between your method and the method of Chen et al (2011) later in the text, too. Their code (and I believe Pollard and Reusch (2002), too) could be tested on the reanalysis data set, too.*

Although it is beyond the scope of this paper, we agree that a full methods-intercomparison study would be useful and we would be happy to collaborate with others on a more detailed intercomparison. In the meantime, the comparison of the figures in our paper (Figs. 11-13) with those in previous discussions of the calendar effect will provide a first-order demonstration of the validity of the various approaches.

*Section 2: You mention the 360-day and 365-day calendar years, it would be worth pointing out that the default code and the shown results are all (?) based on 365-day calendar. Or was it with the actual leap-year calendar, since you worked with reanalysis data.*

We note on line 79 that the Section 2 illustrations are based on a 365-day "noleap" calendar. In Section 3, the CFSR and CMAP data we use are long-term monthly means, also on a 365-day "noleap" calendar, and we will add text to make this point clear to the reader.

*Line 119 and Fig.3: The color scale in the Figure is truncated at 10 Wm−2, but the text describes values up to 35 Wm−2. Please consider an update of the color scale range in the Figure. This may also apply for other figures. Please revisit the color scales and make sure that the range covers the actual value range of the data.*

We carefully designed the color scales we used to allow comparisons within and among the figures, and used diverging (as opposed to a progressive) color palettes that should be accessible to color-deficient viewers (probably ten percent of the readers of this journal). We also used logarithmic class intervals to accommodate the long-tailed distributions of the data. We would therefore like to keep the figures as they are to facilitate that comparison. We have included a "month-length anomalies" .csv file along with the "anomalies" data that

underlie Figs. 3 through 5 and 7 through 9 in the code repository folder
`/data/figure_data/month_length_plots/`.

*Line 127: Here I feel that it is possible to give a more precise value than 10Wm−2, given the range of the color scales in the Figures 7-9.*

We will cite specific values here, and also refer the reader to the appropriate .csv files containing the data.

*Lines 254-269: The description of the effects of the biases in the transient simulation is done well. However, in the discussion section (or here in this paragraph) the authors should discuss the implications for real-world paleoclimate studies. Two things come into my mind: (a) most often paleoclimate studies use seasonal mean data rather than single month data, (b) for proxy model comparisons and interpretation of time series signals one could as: 'Does it matter in the end?' Most likely one does encounter seasonal rather than single month responses in proxies, and processes within the climate system are often analyzed (at least at first) by season rather than single months. I still find it appropriate to present the monthly mean results here, but the practical points of view should also be highlighted.*

We agree that there is more to discuss here, including that the monthly time series (both month-length adjusted and unadjusted) show that the standard "meteorological" seasons are not necessarily the optimal way to aggregate data (September in particular often looks like it belongs more with July and August than with October through November), and months that appear highly correlated over some intervals (e.g. July and June global temperatures from the LGM to the Holocene), become decoupled at other times. We will expand the discussion in Section 3.4 to note some of the additional features revealed by the transient simulations, adding text along the lines of the following:

> There are other interesting patterns in the monthly time series from the transient simulations, some of which are amplified by the calendar effect, and others damped. The monthly time series suggest that the traditional meteorological seasons (i.e., December-February, March-May, June-August, September-November) are not necessarily the optimal way to aggregate data—September time series in Fig. 14 often look like they belong with July and August than with October and November, the traditional other (northern) autumnal months. Figure 14a (North America and Europe), for example, suggests that the July through November time series are similar in overall trends, and that is even more so for the adjusted data (in pink and red). Similarly, months that appear highly correlated over some intervals (e.g. June and July global temperatures from the LGM to the Holocene), become decoupled at other times. The impacts of the calendar effect on temporal trends in transient simulations (Fig. 14), when compounded by the spatial effects (Figs. 11-13), make it even more likely spurious climatic mechanisms could be inferred in analyzing transient simulations than in the simpler time-slice simulations.

We also experimented with a rearrangement of the "columns" in Figs. 1 through 5 and 7 through 9, to place them in "meteorological order" with December on the left and November on the right. This experiment indicated that arranging the data in January to December order (from left to right) was most effective for displaying the data.

As to point (b), a perspective shared by the other referees, it seems that it would be good to add some discussion of an "ideal world" scenario, in which (1) paleoclimatic data would be reconstructed or interpreted in terms of climate variables and indices that are not based on

monthly, seasonal or annual averages, but instead on process-based variables, and (2) climate-model output would be archived on a daily time step, so that those process-based variables could be calculated by post-processing (or even calculated directly within the model).

The comments of all three referees also suggest that a short summary paragraph at the end of Section 3 would be warranted and we will add this summary to the text (as Section 3.5):

> Several observations can be made about the calendar effect, and its potential role in the interpretation of paleoclimatic simulations and comparisons with observations:
>
> - The variations in eccentricity and perihelion over time are large enough to produce differences in the length of (fixed-angular) months that are as large as four or five days, and differences in the beginning and ending times of months on the order of ten days or more (Fig. 1).
>
> - These month-length and beginning and ending date differences are large enough to have noticeable impacts on the location in time of a fixed-length month relative to the solstices, and hence on the insolation receipt during that interval (Figs. 2 through 5). The average insolation (and its difference from present) during a fixed-length month will thus include the effects of the orbital variations on insolation, and the changing month length.
>
> - However, such insolation effects are not offset by the changing insolation itself, but instead can be reinforced or damped (Figs. 7 through 9). (In other words, orbitally related variations in insolation do not "take care" of the calendar-definition issue.)
>
> - The "pure" calendar effects on temperature and precipitation (illustrated by comparing adjusted and non-adjusted data assuming no climate change) are large, and spatially variable, and could easily be mistaken for real paleoclimatic differences (from present).
>
> - The impact of the calendar effect on transient simulations is also large, affecting the timing and phasing of maxima and minima, which, when combined with spatial impacts of the calendar effect, makes transient simulations even more prone to misinterpretation.

*Section 4:*

*I got a bit confused with the technical definition of start day of a year. What is actually determining the first day of a calendar year? Astronomically, vernal equinox for example would make sense, but here I believe it is defined more on technical grounds by the default application and historical developments in climate models? That is, in the present-day calendar we call Jan 1st the first day in a year, which is a certain longitude position of the Earth in its orbit around the sun. Please mention or explain the basis of the definition start day.*

We will revise Section 4.2 to include the application of the Kepler equation approach, and this discussion will make explicit how the beginning of the year was set. This text (along the lines of the following paragraphs) will also explain the counter-intuitive notion of a current

year starting in December of the previous year. We will replace the discussion of the Kutzbach and Gallimore approach by:

Calculation of the length and the beginning, middle and ending (real-number or fractional) days of each month at a particular time is based on an approach for calculating orbital position as a function of time using Kepler's equation:

$$M = E - \varepsilon \cdot \sin(E), \tag{1}$$

where $M$ is the angular position along a circular orbit (referred to by astronomers as the "mean anomaly"), $\varepsilon$ is eccentricity, and $E$ is the "eccentric anomaly" (Curtis, 2014; Eq. 3.14). Given the angular position of the orbiting body (Earth) along the elliptical orbit, $\theta$ (the "true anomaly"), $E$ can be found using the following expression (Curtis, 2014; Eq. 3.13b):

$$E = 2 \tan^{-1}\left(\left((1-\varepsilon)/(1+\varepsilon)\right)^{0.5} \tan(\theta/2)\right) \tag{2}$$

Substituting $E$ into Eq. 1, gives us $M$, and then the time since perihelion is given by

$$t = (M/2\pi)T \tag{3}$$

where $T$ is the orbital period (i.e. the length of the year) (Curtis, 2014; Eq. 3.15).

This expression can be used to determine the "traverse time" or "time-of-flight" of individual days or of segments of the orbit equivalent to the "fixed-angular" definition of months or seasons. Doing so involves determining the traverse times between the vernal equinox and perihelion, between the vernal equinox and January 1 (set the appropriate number of degrees prior to the vernal equinox for a particular calendar), and the angle between perihelion and January 1, and using these values to translate "time since perihelion" to "time since January 1". The "true anomaly" angles along the elliptical orbit ($\theta$) are determined using the "present-day" (e.g. 1950 CE) definitions of the months in different calendars (e.g. January is defined as having 30, 31, and 31 days in calendars with a 360-, 365- or 366-day year, respectively). For example, January in a 365-day year is defined as the arc or "month angle" between 0.0 and 31.0 × (360.0/365.0) degrees. Note that when perihelion is in the Northern Hemisphere winter, the arc may begin after January 1 as a consequence of the occurrence of shorter winter days, and when perihelion is in the Northern Hemisphere summer, the arc may begin before January 1, as a consequence of longer winter days.

We also implemented the approximation approach described by Kutzbach and Gallimore (1988, Appendix A) for calculating month lengths. There were no practical differences between approaches.

*Line 369: the equation is for a variable with physical units of time. On the right-hand side, is the variable phi (an angle in degrees) correct?*

As noted by Referee 2, there is a typo in this equation. Yes, $t$ in Kutzbach and Gallimore's (1988) expression has units of time (days). In response to comments by Referee 2, we will replace this equation with an alternative approach, using a "time-of-flight/traverse time" representation of Kepler's equation that is a bit more intuitive.

*Line 373: "fixed at 80 days after the beginning of the model year" I am not sure if that is clear to all readers, it got me thinking again, what is actually better: To describe the begin of the year as fixed relative to the vernal equinox point, and defined via a longitude angle; or from a modeler's*

*perspective where we are used to think of years starting Jan 1st, and vernal equinox is somehow flexible and can be set to a specific day in the year.*

Yes, there is a choice in where to "anchor" the year, and we followed the custom of using the vernal equinox as the anchor, and January 1[st] is then defined as an angle relative to the equinox.

In either case (defining the start of the year relative to January 1 or the vernal equinox), when perihelion occurs in that segment of the orbit representing the "first month of the year", the amount of time it takes to sweep out that segment will be less than if aphelion occurs in that interval. It turns out that the choice of the fixed-reference point (e.g. the vernal equinox here) has a negligible influence on the calendar effect, because the relative length of months depends only on the shape of the orbit (e.g. on eccentricity, and the time of year of perihelion). The choice does have some effect on the assignment of the middle, beginning and ending dates of each month, once the month lengths have been determined, but that effect is swamped by the month-length changes.

*Line 373: better if you mention here the actual sub-routine that is to be used with the 365-day calendars.*

We will do that (and the name of the subroutine is now simply `monlen()`).

*Section 5: Can you briefly mention which libraries and versions (I think of udunits, netcdf) you used and recommend in connection with the code?*

We will do that in the code repository, in the `/PaleoCalAdjust/f90/` folder `README.md` file

*Figures*

*Figure 2: Can you add the summer solstice as a point in the figure? It could be helpful to have in this figure.*

Yes, we will add the summer solstice point to the x-axis of Figures 1 and 2.

*One question I have though, since there is a little problem:*

*When you compare Dec and Jan the color code switches from plus (blue) to minus (red) colors. At a day in the year 180 days before (after) summer solstice (SS) [in a 360-day calendar] it becomes meaningless to talk about closer and farther relative to SS. An anomaly of one day brings that day closer to SS and at the same time farther away from SS. Can you please explain once more how the reader should interpret the color code in Figure 2? And maybe explain if that has implications on the definition of anomalies shown in Figure 3-5, or not?*

There is a similar abrupt color (i.e. sign) shift between June and July, which may be easier to interpret because it does not involve "wrapping" the calendar between December and January.

Figure 2 essentially maps the displacement of the stack of horizontal bars for individual months, which reflects the changing beginning and end dates of each month. Using 15 ka as an example, perihelion occurs on day 111.87 (relative to January 1), and consequently the months between March and August are shorter than present. That effect in turn moves

the beginning, middle and ending day of the months between April and December earlier in the year. July therefore begins a little over five days earlier than at present—i.e. closer in time to the June solstice. June likewise is displaced earlier in the year, with the beginning of the month 3.36 days farther from the June solstice, and the end a similar number of days closer to the June solstice than at present. Thus the calendar effect arises more from the shifts in the timing (beginning, middle and end) of the months than from changes in their lengths.

We will add material along the lines of the above to the Section 2 discussion of Figs. 1 and 2.

*Figure 3 (and following Fig 4,5):*

*Why is it that all but the Dec months show consistent color coding with Figure 2? In Fig. 3 the December months clearly stand out because they do not vary in terms of anomalies over the past 150,000 years. I also find it inconsistent with the maps in Figure 11.*

Figures 3 to 5 show the calendar effect on insolation at three different latitudes, and that effect (which is longitudinally uniform at each latitude) can be thought of as being composed of the month-length effects superimposed on the time-varying insolation. The amplitude of the calendar effect on insolation in December at 45° N (Fig. 3) only occasionally exceeds the range between -2.0 and +2.0 Wm$^{-2}$ because it is winter in the Northern Hemisphere, and insolation in general is low. Likewise, the calendar effect on insolation at 45° S (Fig. 5) is quite muted in June, which is winter in the Southern Hemisphere.

We will add material similar to the above to the discussion of Figs. 3 to 5 in section 2.

*Anonymous Referee #2*

*This manuscript discusses the problem associated with the usual definition of seasons in climate models, when simulating climates under different orbital (precessional) configurations, a situation rather common in paleoclimatology. The subject is important, sometimes even critical, and too often it is overlooked. So this discussion and more importantly the availability of some useful routines is a significant and valuable contribution. Unfortunately, I have serious critical comments, both on the manuscript itself, but also on the methodology used in the mathematical routines. With these shortcomings I cannot recommend publication of the manuscript in its current state.*

*Major comments*

*– Solving the astronomical problem (§4.2 – lines 350 to 377) The question of the length of seasons is a very classical astronomical problem, since at least the antiquity. It was solved, in the two-body approximation, by Kepler (1609) with the famous first and second laws. The mathematical problem is called "Kepler's equation": M = E – e sinE where M is the mean anomaly (basically time), e is the eccentricity and E is the eccentric anomaly (basically the orbital position). The direct problem (ie. computing M or time for given orbital positions E like equinoxes or solstice for the seasons) is trivial.*

Yes, and we will implement the Kepler-equation approach in the manuscript text, figures, and

accompanying code, to address the Referee's concerns about the use of an approximation by Kutzbach and Gallimore (1988).

*Solving the inverse problem (computing position E as a function of time M) is likely to be part of the climate model code (usually in a routine called "solar". . . but see below), and numerous algorithms have been proposed during the last four centuries to solve it exactly.*

We do not need to solve Kepler's equation in this manner, but we appreciate the Referee's attempt to make the discussion more transparent. We have included a subroutine `kepler_theta(…)`, that given the mean anomaly ($M$), returns the "true anomaly" or angular position along the elliptical orbit, $\theta$.

*In this context, it is extremely disturbing to see a scientific paper based on a very crude approximation of Kepler's equation (manuscript equations 1 and 2) based on a first order expansion in e.*

We initially implemented Kutzbach and Gallimore's (1988) approach because it is relatively transparent and has historical precedence, particularly in the paleoclimate literature. We will replace the Kutzbach and Gallimore (1988) approach in the paper with the Kepler-equation approach suggested by Referee 2, and we have already tested both approaches. Our results using the Kutzbach and Gallimore (1988) approach show good agreement (i.e., to the fourth and fifth decimal places of computed month lengths) with our results using the Kepler-equation approach, indicating that the Kutzbach and Gallimore (1988) approach does provide a good approximation of the results that can be obtained using the Kepler equation.

*To be more explicit (see standard textbooks. . .): M =(2π/T) t E = 2 Arctan[ sqrt((1-e)/(1+e)) tan(v/2) ] v = λ – ÏU˝ + π (T = 1 year, t = time from perihelion, v = angular orbital position from perihelion, λ = angular orbital position from some reference, usually spring equinox, ÏU˝ = climatic precession, or angular orbital position of perihelion versus a reference, usually vernal point, π = 180◦ = spring equinox vs. vernal point) So, for given orbital parameters (precession ÏU˝ ; eccentricity e) it is rather simple to compute time t as a function of λ using Kepler's equation. I do not understand the need for a first order approximation in e when equipped with a computer. . . Where does it come from ? Kutzbach and Gallimore (1988); citing Symon (1964). . . probably citing someone who had no pocket calculator and found it useful to write 1st order approximations. This is no more relevant and, I think, not acceptable today: Arctan, tan, Sqrt are usually available in most computer languages. . .*

We modified the month-length program by adding what is sometimes referred to as "Kepler's 'time-of-flight' or 'traverse-time' equation" as discussed in Curtis (2014) *Orbital Mechanics for Engineering Students* (Ch. 3, Orbital position as a function of time). Curtis's (2014) discussion and the Referee's parallel one another (when allowance is made for the symbol corruption above).

*– The same kind of unacceptable "pedestrian" procedure is used (line 374) for solving the equation sin(x)=-1 Using standard mathematics, the solution is x = -π/2 The corresponding algorithm described in the manuscript appears, to say the least, a bit awkward: Ân´ Select φp that minimizes {-1-sin((2π/360)(φ φp))} Âz˙ I prefer to write it more simply as Ân´ φp = φ + π/2 (or in degrees φp = φ + 90◦) Âz˙ and would not use a computer minimization routine for that. . .*

We will replace this section of text describing the Kutzbach and Gallimore (1988) approach with text describing the Kepler-equation approach as suggested by Referee 2.

*Overall, I do not understand why the whole procedure is so complex,  and divided  in five computer procedures (Step 1 to 5). It is based on unnecessary approximations and on very awkward procedures like ". . .advancing along the orbit at 0.001-day increments. . .".*

As for the division of the code into multiple steps, we did that for transparency.  It is certainly the case that the steps could be concatenated, but as written, they allow an interested user to see the incremental steps.  A reader might note that much of the code is devoted to accommodating the range of calendars (in the CF sense) used in different models.

We will replace the description of the Kutzbach and Gallimore (1988) approach for calculating month lengths (including the text describing the procedural steps) with a description of the calculation of month-lengths using the Kepler-equation approach as suggested by Referee 2.

*As a result, I have some severe reservations on the relevance or validity of the code. As explained above, the time between 2 orbital positions is just a simple application of Kepler's equation: t2 - t1 =(T/2π)(M2 - M1) = (T/2π)( E2 − e sinE2 − E1 + e sinE1 ) = ... with (E1, E2) functions of orbital positions (λ1, λ2) as explained above. This should stand in 2 or 3 lines of computer code, no more.*

We have implemented the Kepler-equation approach suggested by Referee 2.  As noted above, the Kutzbach and Gallimore (1988) approach and the Kepler-equation based approach yield month-length values that differ only in the fourth or fifth decimal places and thus our discussion of the results remain largely unchanged.  We have redone all of the figures using the Kepler-equation approach results, and those figures are included in the code repository, allowing comparisons with the Kutzbach and Gallimore (1988) approach (as displayed in the original figures in the GMDD discussion paper).

*– A large part of the manuscript is about describing the "impact of the calendar effect" (§3 – lines 133 to 300). Most of this has been already described and discussed in numerous papers. I believe this part is far too long and far too descriptive. Again, most of this is known since quite a long time: geologists in the XIXth and early XXth centuries were discussing the astronomical forcing not in radiative terms (W/m2) but in terms of season duration (number of days of winter versus summer). This "calendar effect" is in fact at the heart of the precessional forcing. The key point is not so much its existence or its relevance, but how to deal with it in order to interpret model results in seasonal terms.*

Despite this long history, it is surprising that recognition of the calendar impact and application of ways of dealing with it is not a routine part of paleoclimatic analysis.  A novel contribution of our paper is the demonstration of the "pure" calendar effects on insolation, temperature and precipitation, which in previous papers have been examined using long-term mean differences (between "paleo" and control simulations), which combine calendar and climatic-change effects.

*- Lines 320-348. I do not understand why the "mean-preserving" interpolation does not preserve exactly the mean. The fact that the error is small (but can be as big as -0.12 mm/day) is not reassuring. Where does the error come from? If there is no other mathematical treatment that the "mean-preserving interpolation", then I expect the mean to be exactly preserved, not approximately.*

As is also the case with the parabolic spline interpolation of Pollard and Reusch (2002), Epstein's (1991) approach can occasionally produce overshoots that are physically impossible. For practical reasons, variables like precipitation are therefore "clamped" at zero, which introduces the error. The example we presented for the Indian Ocean was the worst-case example. Figure 15 shows that interpolation errors of this size are rare. We will add summary statistics to the figure legend to better describe these errors for the reader.

*Minor comments*

*line 30-31 : Ân´ larger number of days Âz˙      Should be the opposite.*

Yes. We will change "larger" to "smaller".

*Please state somewhere in the introduction that a fixed angular month is precisely 30◦. Stating that a "fixed-angular month corresponds to a fixed number of degrees" is not sufficient. It should also be stated that all "fixed-angular month" are equal, and therefore equivalent to 360◦/12. Without such a clear statement, the definition of a "fixed-angular month" is very ambiguous.*

Fixed-angular months of 30 degrees are a special case. For a 365-day "noleap" calendar, January, as an example, has a fixed-angular definition of $31.0 \times (360.0 / 365.0)$ degrees (see text discussion in Sections 1 and 4.2).

*line 360 : Ân´ To calculate the orbital parameters . . . we adapted a set of programs . . . https://data.giss.nasa.gov/ar5/solar.html Âz˙ This link is broken. (I am pretty sure this solves Kepler's equation )*

Yes, that link is now broken. However, the web page is available on the Wayback Machine at: https://web.archive.org/web/20150920211936/http://data.giss.nasa.gov/ar5/solar.html (accessed 2019-01-29) and we have added this URL to the code. The modified orbital parameter programs we used here are in the code repository accompanying the manuscript.

*line 369 : equation (2) there is a missing index p in the second cosine (see the original ref).*

We will replace this equation with text describing the Kepler-equation approach.

*Anonymous Referee #3*

*The authors present the software which can convert the standard output of PMIP models to the new, celestial, calendar. The authors believe that such a calendar is better than the standard (fixed day) one. This is my main disagreement with the authors and other reviewers. I simply do not believe that the celestial calendar is better (or worse) than the standard one.*

We are not arguing that a celestial (or angular) calendar is inherently "better." As we describe in the introduction, the issue we are addressing is that paleoclimate analyses frequently compare data (i.e., simulated and/or observed data) for two different time periods assuming a fixed-length definition of months. This approach can introduce a calendar effect in the results that can mimic observed paleoclimatic changes. At one time it was assumed that the calendar effect was not large enough to warrant explicit consideration, while

elsewhere attempts were made to adjust for the calendar effect using relatively simple approaches (e.g. Harrison et al., 2014). However, the amplitude of the calendar effect along with the tendency for the resulting map patterns to resemble plausible paleoclimatic patterns, and the impact of the effect on phasing in transient experiments—all effects that indeed have been previously demonstrated by others—have convinced us that it is important to address the calendar effect in paleoclimatic analyses. Referee 3's comments are more of a discussion of the philosophy of how we do paleo science involving data and models, than a technical review, and we think that this discussion will indeed help advance that science.

*The authors begin their paper from the statement that "there are two ways of defining month or seasons" (p. 1). This is of course not true since there is a myriad of ways to define months and seasons. Julian and Gregorian calendars are obvious examples.*

Julian and Gregorian calendars are both "fixed-length" calendars (too fixed, in the case of the Julian). We will modify the text of Section 1 to make clear to the reader how we are defining fixed-length and fixed-angular calendars.

*For paleoclimate applications, there are many other options. For example, one can set the summer solstice to 22 June instead of setting vernal equinox to March 21, as is required by the PMIP protocols. In fact, fixing of the summer solstice would be more reasonable, at least for the Northern Hemisphere.*

Jousaumme and Braconnot (1997) discussed the choice of the "reference point" and show (their Fig. 2) that it has an impact on calculated insolation and long-term differences. However, the choice of a fixed reference point does not change the shape of the orbit, and ultimately the time it takes Earth to transit different segments of it. As we note, and Referee 3 acknowledges, the vernal equinox is required for paleoclimate simulations using the CMIP/PMIP protocols, which is one of the reasons we have used it here.

*While for the present day the calendar has absolute meaning since observational climate data used for model validations are aggregated according to the "official" months, for the analysis of past climate simulations in principle one is free which calendar to use. For model intercomparison, the only important requirement is that all models should use the same calendar.*

Inspection of the CMIP5/PMIP3 model output shows that in practice, a variety of different calendars have been used by different modeling groups, and we realize there may not be sufficient flexibility in the models themselves to allow selection of a specific calendar that could be used by all modeling groups. But even if the same calendar was used, calendar effects would still pervade model intercomparison results. In Section 3.1, for example, we show that the "pure" calendar effect on temperature is dependent on the amplitude and phase of the seasonal cycle, and these in turn are dependent on a model's spatial resolution and its influence on model orography. So even in the uncomplicated (by paleo data) case of model-only intercomparisons, calendar effects will still be an issue.

We will include a short discussion of the calendar effect on model-only intercomparisons in the second paragraph of the introduction, and at the end of Section 3.1.

*For comparison with paleoclimate proxies, any calendar is of limited use because the calendar is human invention and Nature has no idea about seasons or months. Therefore model/data comparison cannot be improved by choosing the "right" calendar. To the contrary, proper*

*model/data comparison requires abandoning of any calendar and using climate characteristics which are independent of the choice of the calendar.*

*Of course, ideally observed proxy records should be compared with the simulated ones.*

We agree. Using fossil-pollen data as an example paleoclimatic data source (Bartlein et al., 2011; Harrison et al., 2016), one can observe a slow evolution toward reconstructing more biologically or physically based variables such as growing degree days or plant-available moisture, in addition to conventional monthly or seasonal meteorological variables. Such variables make no prior demands on the definition of months or seasons (but they do require daily data). Similarly, "forward models" and "proxy-system models" are evolving to allow for direct comparison of paleoclimatic evidence and model output. However, there is still a considerable wealth of existing reconstructions of conventionally defined variables, and it is important to account for calendar effects when using these data.

We will include a statement like this in Section 6.

*Let's consider the advantages of using celestial calendar compare to the standard (fixed-day) one. For two special orbital configurations, namely, when summer solstice coincides with perihelion or aphelian ("warm" and "cold" orbits respectively) celestial calendar has one obvious advantage –the maxima and minima of insolation will always occur at the same days (90 and 270 days of celestial calendar) while under large eccentricity when using the standard calendar, the summer solstice (and maxima/minima of insolation) can deviate from 22 June by +-5 days.*

The calendar effect at 116 ka comes close to illustrating the "warm orbit" scenario (perihelion occurs 3 days after the June solstice, and eccentricity is relatively high).

*However, for the two "representative" months, January and July, the differences between the standard and celestial calendars (as shown by numerous figures in the Bartlein and Shafer manuscript) are rather small.*

The calendar effects on temperature for those two months at 116 ka are indeed small, but those on July precipitation are less so, and as Referee 3 notes below, they grow from July into November.

*These differences increase significantly during the transition months (August- November). Which of two calendars is better for these months? The simple answer is NONE because these months exist only in our imagination and I cannot see any sense in comparison, for example, September temperatures at present and 127 000 years ago. However, other workers may disagree with me and want to analyze climate change during spring or autumn.*

Other researchers indeed might want to look at September conditions, if they were interested in, say, Northern Hemisphere monsoon-related precipitation variations, or the initiation of annual sea-ice growth.

*In this case, they have to realize that for these months, the celestial calendar has a serious problem even compare to the standard (fixed-day) one because it corrupts the most fundamental characteristics of the real world – time. For a high eccentricity, the days in the celestial calendar can be 10% shorter or longer than the real ones and, as the result, the beginning for example of celestial "October" can move back and forward compare to the summer solstice by more than 10 "real" days (Fig.2). At the same time, the internal time scales*

*of the climate system do not depend on the orbital parameters and therefore the time lags between insolation and climate characteristics remain nearly constant in the real time, not in the celestial "days". Thus using of celestial calendar corrupts the physics of climate.*

When using a celestial or angular calendar, the rate of rotation of the Earth does not change, nor does the length of a sidereal day, just the length of the Earth's orbit that is used to define months and seasons, which in turn determines its insolation "exposure." We find it difficult to imagine how relatively simple adjustments in the intervals over which a climate variable is averaged or accumulated corrupts physics in any way.

*It is noteworthy that in the paper by Kutzbach and Gallimore (1988) cited in the manuscript and where celestial calendar has been used, Kutzbach and Gallimore explicitly stated (page 820, first para) about the celestial calendar:*

*"The procedure, however, is mainly applicable to climate experiments that prescribe ocean and sea ice conditions, i.e., climate systems not having interactive components with significantly different lags in response to solar forcing".*

*Thus Kutzbach and Gallimore already 30 years ago clearly realized that corruption of absolute time is a serious problem. Surprisingly, modern authors seem to be unaware of this problem.*

In the quotation above, Kutzbach and Gallimore (1988) were referring to a weighting procedure for calculating annual means that was applied in an earlier paper (Kutzbach and Otto-Bliesner, 1982, p. 1178), and not to the approach for calendar adjustments Kutzbach and Gallimore developed in their 1988 paper, nor to any notion of the "corruption of absolute time." The full quotation is:

> "Kutzbach and Otto-Bliesner (1982) devised a strategy for comparing averages obtained from the 9000-year B.P. and control (0-year B.P.) experiments performed with an earlier version of the A model that prescribed constant orbital speed. Their approach was to compute annual averages by proper weighting of seasonal or monthly averages according to the number of days in the celestial season (90°) or celestial month (30°). The procedure, however, is mainly applicable to climate experiments that prescribe ocean and sea ice conditions, i.e., climate systems not having interactive components with significantly different lags in response to solar forcing." (Kutzbach and Gallimore, 1988, pp. 819-820)

*Above I only discussed the situation with two very specific orbital configurations –when summer equinox occurs in perihelion or aphelion (like that at 126 ka or 116 ka). What about an arbitrary Earth's orbit? For any arbitrary orbit, the only advantage of the celestial calendar disappears because maxima and minima of insolation at different latitudes do not coincide anymore with the solstices and can deviate from them by up to one week, i.e. as much as they can deviate from 22 June and 22 December in the standard calendar.*

*By saying that, I want to make it clear that I am not against using several different calendars. This at least helps to understand that at the orbital time scales, things like "spring" or "October" do not have any meaning. But to be useful, the manuscript under consideration should not make false impression that it presents The Solution for the Calendar Problem and that Celestial Calendar is the right one. I believe, the manuscript requires a thorough discussion of problems and limitations of any calendar applied to the analysis of model results.*

We agree (except for the notion that we have made "false impression[s]"). And, as noted above, we are not arguing that a celestial calendar is "better" than other calendars but rather that using a calendar with fixed-angular months can help to account for calendar effects in paleoclimate analyses. In addition, we also maintain that by a) reviewing previous work, b) refocusing the source of the calendar effect discussion from a simple short-month/long-month explanation (Fig. 1) to a location along the orbit relative to the solstices one (Fig. 2), c) illustrating the "pure" (as opposed to confounded by actual paleoclimatic change) effects on snapshot and transient experiments (Figs. 11-14), and d) providing a practical method of adjusting for the calendar effect, that we have exceeded the threshold necessary for a "thorough discussion."

**References**

Bartlein, P. J., Harrison, S. P., Brewer, S., Connor, S., Davis, B. A. S., Gajewski, K., Guiot, J., Harrison-Prentice, T. I., Henderson, A., Peyron, O., Prentice, I. C., Scholze, M., Seppa, H., Shuman, B., Sugita, S., Thompson, R. S., Viau, A. E., Williams, J., and Wu, H.: Pollen-based continental climate reconstructions at 6 and 21 ka: a global synthesis, Climate Dynamics, 37, 775-802, https://doi.org/10.1007/s00382-010-0904-1, 2011.

Chen, G.-S., Kutzbach, J. E., Gallimore, R., and Liu, Z.: Calendar effect on phase study in paleoclimate transient simulation with orbital forcing, Clim. Dynam., 37, 1949-1960, https://doi.org/10.1007/s00382-010-0944-6, 2011.

Curtis, H. D.: Orbital position as a function of time, in: Orbital Mechanics for Engineering Students, 3rd edition, Elsevier, Amsterdam, 145-186, 2014.

Epstein, E. S.: On obtaining daily climatological values from monthly means, J. Climate, 4, 365-368, https://doi.org/10.1175/1520-0442(1991)004<0365:OODCVF>2.0.CO;2, 1991.

Harrison, S. P., Bartlein, P. J., Brewer, S., Prentice, I. C., Boyd, M., Hessler, I., Holmgren, K., Izumi, K., and Willis, K.: Climate model benchmarking with glacial and mid-Holocene climates, Climate Dynamics, 43, 671-688, https://doi.org/10.1007/s00382-013-1922-6, 2014.

Harrison, S. P., Bartlein, P. J., and Prentice, I. C.: What have we learnt from palaeoclimate simulations?, Journal of Quaternary Science, 31, 363-385, https://doi.org/10.1002/jqs.2842, 2016.

Joussaume, S. and Braconnot, P.: Sensitivity of paleoclimate simulation results to season definitions, J. Geophys. Res.-Atmos., 102, 1943-1956, https://doi.org/10.1029/96JD01989, 1997.

Kepler, J.: New Astronomy (*Astronomia Nova*, 1609), translated from the Latin by W. H. Donahue, Cambridge University Press, Cambridge, England, 681 pp., 1992.

Kutzbach, J. E. and Gallimore, R. G.: Sensitivity of a coupled atmosphere/mixed layer ocean model to changes in orbital forcing at 9000 years B.P., J. Geophys. Res.-Atmos., 93, 803-821, https://doi.org/10.1029/JD093iD01p00803, 1988.

Kutzbach, J. E. and Otto-Bliesner, B. L.: The sensitivity of the African-Asian monsoonal climate to orbital parameter changes for 9000 years B.P. in a low-resolution general

circulation model, J. Atmos. Sci., 39, 1177-1188, https://doi.org/10.1175/1520-0469(1982)039<1177:TSOTAA>2.0.CO;2, 1982.

Pollard, D. and Reusch, D. B.: A calendar conversion method for monthly mean paleoclimate model output with orbital forcing, J. Geophys. Res.-Atmos., 107, ACL 3-1-ACL 3-7, https://doi.org/10.1029/2002JD002126, 2002.

---

## Author Response (AR2)

Authors' Response

We thank the referee for their thoughtful comments and respond to their individual comments below.

In the following text, the referees' comments are in italic font followed by our authors' responses in regular font.

*Submitted on 21 May 2019*
*Anonymous Referee #2*

*In this revised version of their manuscript, the authors have answered satisfactorily my main concerns. I have therefore no more major comment on this paper. Overall the paper is well written and informative and I therefore recommend its publication. I have nevertheless two minor comments below that could be helpful to improve the final manuscript.*

*Minor comments :*

*1 - Line 25 and line 27 : « number of sidereal days » I don't think that this number is what the authors have in mind. A sidereal day is a rotation with respect to the stars, while a synodic day is a rotation with respect to the sun, and an actual day, based on Universal Time is defined by 24 hours (There are about 366,25 sidereal days in one year, each sidereal day having a duration of 23h 56min and 4 secs). Please remove « sidereal », and use « day » in its usual meaning (= 24h).*

We deleted "sidereal" and use "day" (lines 25 and 27).

*2 - Line 387-390 : This concerns my previous comment about « mean-preserving » interpolation. I now understand where the errors come from, but it is still not explained in the manuscript. It is written « … the Epstein approach can create interpolated curves that are wavy… » : this is not helpful to understand why the monthly mean of the interpolation is not equal to the original monthly mean. The authors need to add that precipitations are « clamped at zero » to remain physically relevant, therefore the occurrence of small errors in some places. Without such an explanation, the manuscript remains awkward since the « mean preserving method » does not preserves the mean.*

We added the following text to the first paragraph of Section 4.1 (lines 400-404 in the "track changes" version (line 378-382 in the "changes accepted" version):

[revised manuscript text omitted]